# CRYSTALAGENT: TOWARDS AUTONOMOUS CRYSTAL GENERATION VIA A FOUR-STAGE AGENT

## ABSTRACT

Recent advances in large language models (LLMs) have demonstrated remarkable generalization capabilities across diverse domains, and recent studies have begun to explore their application in crystal generation. Nevertheless, most of these approaches rely heavily on extensive fine-tuning with large-scale datasets, which often limits their adaptability and generality when applied to real-world crystal discovery. To overcome these limitations, we propose CrystalAgent, an LLM-based agent that eliminates the need for additional training and adapts flexibly to diverse crystal discovery scenarios. Specifically, we decompose the crystal generation process into four key stages: Extract, Retrieval, Generation, and Optimization. The Extract stage involves extracting crystal design constraints from user inputs. In the Retrieval stage, based on the extracted constraints, the system automatically selects few-shot examples from the database to inform subsequent processes. The Generation stage leverages LLMs to generate crystal structures by learning atomic distribution patterns from the selected examples, and the Optimization stage refines the generated structure by using crystal structure optimization tools and energy evaluation tools to select the optimal structure as the final output. Extensive experiments across various crystal generation tasks highlight the flexibility, controllability, and versatility of our framework, underscoring the substantial potential of LLM agents in automating the generation of crystal materials and advancing the field of materials discovery.

## 1 INTRODUCTION

Crystal materials play a central role in modern science and technology, driving advances in batteries, semiconductors, catalysis, and pharmaceuticals (Ou et al., 2022; Li et al., 2025; Hu et al., 2025b; Zhou et al., 2025). Their physical and chemical properties are governed directly by their atomic structures, making the discovery of novel crystal structures a central problem in materials science. However, the vast chemical design space and the intricate relationship between composition, structure, and property make this task profoundly challenging. Traditional search strategies combined with first-principles calculations have provided an essential foundation for crystal discovery. However, their high computational cost still limits scalability and prevents efficient exploration of the chemical space.

In recent years, machine learning methods have been introduced to accelerate crystal discovery by directly learning patterns from large databases of known structures. For example, equivariant graph neural networks (EGNN) (Satorras et al., 2021) have been developed to encode translational invariance and crystalline symmetry, while generative frameworks such as variational autoencoders (Kingma & Welling, 2014) and diffusion models (Ho et al., 2020) have enabled efficient sampling and property-conditioned crystal generation (Xie et al., 2022; Jiao et al., 2023; 2024).

The rapid development of large language models (LLMs) have begun to reshape the landscape of scientific discovery. Beyond their well-established applications in natural language processing, LLMs are increasingly integrated into the scientific workflow, supporting tasks such as retrosynthesis prediction, drug design, and protein structure generation (Zhong et al., 2024; Wang et al., 2025; Xiao et al., 2025). Motivated by the recent progress of LLMs in diverse domains, researchers have started to investigate their application in crystal discovery. CrystaLLM (Antunes et al., 2024) trains an autoregressive model over CIF text to generate plausible crystal structures. Mat2Seq (Yan et al.,

2024) proposes an invariant sequence representation that encodes crystal geometry in a unique, symmetry-aware form; CrystalTextLLM (Gruver et al., 2024), built on LLaMA-2 (Touvron et al., 2023), enables unconditional, conditional, and infilling generation of crystal structures. MatExpert (Ding et al., 2025) augments LLMs with retrieval and reasoning to emulate expert workflows.

In spite of notable breakthroughs, both deep learning approaches and fine-tuned LLM-based methods still face an important practical limitation. Their effectiveness often hinges on task-specific supervised training on large, curated structural datasets, which is difficult to guarantee in crystallography where experimentally verified structures remain relatively scarce and many available entries are derived from theoretical calculations. As a result, adapting such models to new crystal design tasks or unexplored chemical spaces often requires additional data collection and retraining, which limits their flexibility in practical discovery scenarios.

To bridge the gap, we introduce CrystalAgent, an LLM-based agent that eliminates the need for additional training and adapts flexibly to diverse crystal discovery scenarios. The framework consists of four key stages: (1) **Extract** converts natural language format user queries into structured crystal design constraints. (2) **Retrieval** automatically retrieves a set of structurally similar crystals from the database based on the design constraints, which guide the subsequent structure generation process. (3) **Generation** leverages the in-context learning capabilities of LLMs to generate target crystal structures guided by retrieved examples, while also performing automatic validation of the generated results. (4) **Optimization** relaxes the generated candidates and selects the lowest predicted energy-per-atom structure of the same composition as the final output. Our main **contributions** are summarized as follows:

- ⋆ We introduce CrystalAgent, an LLM-based agent that eliminates the need for additional training and leverages in-context learning to enable flexible crystal generation;
- ⋆ We propose a constraint-driven paradigm that formalizes user intent into crystal design constraints, providing a unified interface for design requirements and enabling adaptive retrieval of similar examples to guide LLM-based crystal generation;
- ⋆ Extensive experiments across diverse crystal generation tasks demonstrate the effectiveness of the framework, highlighting its flexibility, controllability, and potential to advance automated materials discovery.

## 2 PRELIMINARY

A crystal structure can be defined as a material in which constituent particles are arranged in a highly ordered, periodically repeating pattern that extends across three spatial dimensions. The unit cell is the fundamental repeating unit that defines and reproduces the periodic arrangement throughout the entire crystal. In the formal description of a unit cell containing $N$ atoms, the structural information can be represented by the triplet $\mathcal{M} = (\mathbf{A}, \mathbf{X}, \mathbf{L})$. Formally, $\mathbf{A} = [\boldsymbol{a}_1, \boldsymbol{a}_2, \ldots, \boldsymbol{a}_N]^{\mathrm{T}} \in \mathbb{R}^{N \times K}$ denotes the one-hot encoding of atomic species, where $K$ corresponds to the total number of distinct atomic types. The matrix $\mathbf{X} = [\boldsymbol{x}_1, \boldsymbol{x}_2, \ldots, \boldsymbol{x}_N]^{\mathrm{T}} \in \mathbb{R}^{N \times 3}$ specifies the Cartesian coordinates of the atoms within the unit cell. The lattice periodicity is captured by $\mathbf{L} = [\boldsymbol{l}_1, \boldsymbol{l}_2, \boldsymbol{l}_3]^{\mathrm{T}} \in \mathbb{R}^{3 \times 3}$, which encodes the primitive translation vectors of the crystal. On this basis, the infinite crystal structure is rigorously defined as the periodic replication of the unit cell under the action of the translational group generated by $\mathbf{L}$:

$$\{(\boldsymbol{a}_i', \boldsymbol{x}_i') | \boldsymbol{a}_i' = \boldsymbol{a}_i, \boldsymbol{x}_i' = \boldsymbol{x}_i + \boldsymbol{k}\mathbf{L}, \forall \boldsymbol{k} \in \mathbb{Z}^{1 \times 3}\}, \tag{1}$$

where the components of the integer vector $\boldsymbol{k}$ specify the integral coefficients of three-dimensional translations along the corresponding lattice basis vectors defined by $\mathbf{L}$.

To capture the inherent periodicity of a crystal structure, it is often more appropriate to express atomic positions in terms of the lattice vectors $(\boldsymbol{l}_1, \boldsymbol{l}_2, \boldsymbol{l}_3)$ rather than with respect to the conventional orthogonal Cartesian basis. Under this formulation, a Cartesian coordinate $\boldsymbol{x} = \sum_{i=1}^{3} f_i \boldsymbol{l}_i$ is equivalently represented by the fractional coordinate vector $\boldsymbol{f} = [f_1, f_2, f_3] \in [0, 1)^3$. In this study, we adopt the fractional coordinate framework and represent the crystal as $\mathcal{M} = (\mathbf{A}, \mathbf{F}, \mathbf{L})$, where $\mathbf{F} \in [0, 1)^{N \times 3}$ contains the fractional coordinates of all atoms in the unit cell. This work focuses on three primary tasks:

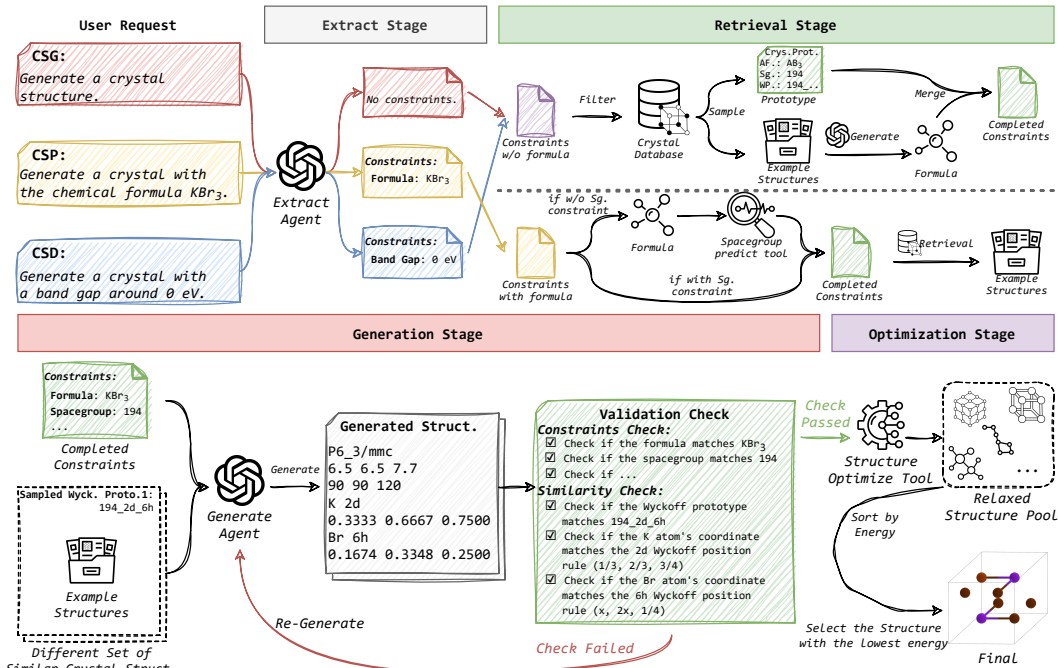

Figure 1: The workflow of CrystalAgent for crystal generation.

**Crystal Structure Generation** (CSG) is concerned with the unconditional generation of crystal structures without any prior specification of chemical composition or constraints. Formally, the objective can be expressed as learning a distribution $p(\mathcal{M})$ over the space of crystal structures. The model distribution $p_\theta(\mathcal{M})$ is required to approximate the empirical distribution $p_{data}(\mathcal{M})$ obtained from the training dataset.

**Crystal Structure Prediction** (CSP) is concerned with determining the stable crystal structure corresponding to a given chemical composition. Formally, for a specified composition $C$, the problem can be expressed as $\hat{\mathcal{M}} = \arg\max_{\mathcal{M}} p_\theta(\mathcal{M}|C)$, where $p_\theta(\mathcal{M}|C)$ is the conditional distribution of crystal structures given composition $C$.

**Crystal Structure Design** (CSD) is the inverse problem of crystal property prediction, wherein the objective is to construct or optimize a crystal structure $\mathcal{M}$ such that it satisfies target physical or chemical constraints. The generative process can be described as $\mathcal{M} \sim p_\theta(\mathcal{M}|\mathcal{C})$, where $\mathcal{C}$ denotes the set of desired constraints.

## 3  METHODOLOGY

In this section, we introduce CrystalAgent, as illustrated in the Figure 1. First, we discuss how crystallographic principles are used to encode crystal structures in a way that facilitates the learning of crystal structure patterns by LLMs. We then provide a detailed description of the four stages of CrystalAgent: Extract, Retrieval, Generation, and Optimization. In the Extract stage, we employ LLMs to process user's natural language input, extracting the crystal design constraints for the desired crystal structure. In the Retrieval stage, we design a process in which the agent identifies and supplements any missing crystal design constraints. It then automatically samples a set of crystal structure examples from a database that adhere to the specified constraints. These selected examples serve as a basis for guiding the subsequent crystal structure generation process. During the Generation stage, we guide the LLM to learn atomic distribution patterns from a limited set of crystal structure examples, enabling it to generate target crystals with similar structures. The generated structures are then validated against the crystal design constraints and the atomic distribution patterns of the example structures. Finally, in the Optimization stage, we use crystal structure optimization tools to refine the structures generated by the LLM and select the lowest-energy candidate as the final output.

## 3.1 SYMMETRY-BASED ENCODING OF CRYSTAL STRUCTURES

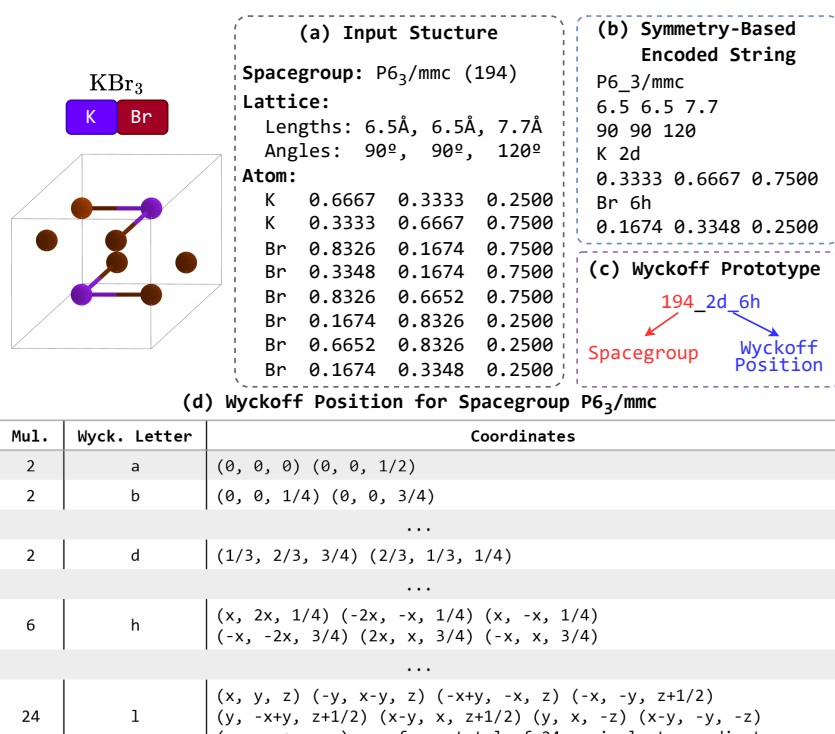

Figure 2: Illustration of Symmetry-Based Encoded crystal structure. (a) illustrates the example $KBr_3$ crystal structure, (b) displays the Symmetry-Based Encoded crystal structure, (c) presents the Wyckoff Prototype proposed for distinguishing crystal structures, and (d) shows a subset of the Wyckoff positions for the space group $P6_3/mmc$.

As described in the Preliminary section, crystal structures are typically composed of three components: atomic species $\mathbf{A}$, lattice constants $\mathbf{L}$, and atomic coordinates $\mathbf{X}$. Through the combination of these three elements, we can fully describe the infinite arrangement of unit cells in three-dimensional space. However, in crystallography, the symmetry of the crystal is also a crucial feature for understanding and classifying crystal structures. Crystal symmetry refers to the operations, such as rotation, reflection, and translation, that can leave the crystal structure invariant. Space groups, as a mathematical method to describe all symmetry operations in a crystal, can classify all crystal structures into 230 distinct space groups. For each space group, the positions of atoms within the unit cell are subject to different symmetry constraints. When a specific space group is given, Wyckoff positions are used to describe the atomic positions within the unit cell and assign them to symmetry-equivalent points determined by the space group's symmetry operations. The numbering of these positions is indicated by their multiplicity and letter designation. For example, for the $KBr_3$ crystal, its space group is $P6_3/mmc$ (194), the positions of the K and Br atoms in the unit cell can be represented by Wyckoff coordinates: K at 2d and Br at 6h, as shown in Figure 2.

Based on the aforementioned crystallographic concepts, each crystal structure can be represented by the following components:

- *Space group symbol*: The internationally recognized symbol representing the space group.
- *Lattice constants*: The essential geometric parameters of the unit cell, including the lengths of the lattice vectors $(a, b, c)$ and the angles between them $(\alpha, \beta, \gamma)$.
- *Atomic positions*: The atomic species, Wyckoff positions, and fractional three-dimensional coordinates of the atoms in the unit cell, along with the corresponding constraints based on the symmetry operations of the assigned space group.

## 3.2 EXTRACT STAGE FOR CRYSTAL DESIGN CONSTRAINTS EXTRACTION

In crystal generation tasks, users may impose various constraints on the properties of the generated crystals. For instance, users might request that the crystals have specific chemical formulas, certain properties, or follow a particular pattern in their chemical composition, potentially including specified elements. To address these requirements, we propose a Crystal Design Constraint Extraction Agent that extracts diverse user intents expressed in natural language and transforms them into crystal design constraints. Given the user query $p_{user}$ and the predefined system prompt $p_{sys\_ex}$, the agent formulates the intended task as follows:

$$c_f, c_{sg}, c_p, c_{af}, c_e = \text{LLM}(p_{sys\_ex}, p_{user}), \tag{2}$$

In our agent architecture, the intent recognition and crystal design constraint generation process produces five types of crystal design constraints, each defined as follows:

- *Chemical formula* $c_f$: The empirical chemical expression constraint of the crystal.
- *Space group number* $c_{sg}$: The space group number constraint of the crystal.
- *Crystal properties* $c_p$: Constraints on the physical or chemical properties of the crystal, such as band gap, formation energy, etc.
- *Anonymized formula* $c_{af}$: The anonymized formula constraint of the crystal, where unique species are arranged in order of increasing amounts and assigned ascending letters, useful for prototyping.
- *Elements* $c_e$: The specific elements that should be included in the crystal.

It is important to note that the types of constraints extracted from user input may vary depending on the type of the crystal generation task. Specifically, for the CSG task, no constraints are typically provided by the user, as the goal is to explore potential structures without specific restrictions. In the CSP task, only the chemical formula constraint $c_f$ is typically provided, as the user aims to predict crystals with a particular composition. In contrast, the CSD task allows users to impose any combination of the five types of constraints outlined earlier, thereby enabling the targeted design of crystals with specific properties, compositions, or symmetries.

## 3.3 RETRIEVAL STAGE FOR FEW-SHOT DEMONSTRATION SELECTION

To fully leverage the context generation capabilities of LLMs, it is essential to retrieve a set of structurally similar crystal structures that align with the user's intent from the database as demonstrations. To achieve this, we propose a Constraints-based Similar Structure Retrieval Agent, which automatically retrieves a set of structurally similar crystal structures from the structure database based on crystal design constraints.

Specifically, we define the similarity of crystal structures from two perspectives: the anonymized chemical formula and the Wyckoff prototype. The anonymized chemical formula constrains the proportions of different elements in the structure, while the Wyckoff prototype is a concept we propose for classifying crystal structures. It consists of two components: the space group and all Wyckoff positions within the lattice. As illustrated in the Figure 2 (c), we concatenate the space group number with the symbols of all Wyckoff positions arranged in ascending order in the lattice to form the Wyckoff prototype.

To retrieve a set of structurally similar crystal structures based on the given crystal design constraints, it is necessary to complete any missing parts of the constraints. If the chemical formula constraint cannot be inferred from the user's input, the first step is to retrieve a set of crystal structures that are structurally similar, sharing the same anonymized chemical formula and Wyckoff prototype, and satisfying other provided constraints, such as element composition. Subsequently, a LLM is employed to generate a plausible chemical formula based on the retrieved crystal structures, thereby completing the chemical formula constraint. The generated formula is then validated by ensuring electroneutrality and verifying that it matches the anonymized chemical expression of the retrieved example crystal structures. Once these conditions are met, the chemical formula and space group constraints are completed, and a set of structurally similar crystal structures that satisfy the constraints is obtained. If the user has provided the chemical formula constraint, the completion process depends on whether the space group constraint is also provided. If the space group constraint is not provided, a space group prediction tool (Li et al., 2021) is used to analyze the symmetry of the crystal structure corresponding to the provided chemical formula, thereby determining the space

group. Following this, the database is filtered using both the chemical formula and space group constraints. From the filtered results, a group of crystal structures with the same Wyckoff prototype is randomly selected to serve as the example crystal structure.

In summary, the output of the Constraints-based Similar Structure Retrieval Agent consists of the completed chemical formula and space group constraints, together with other relevant crystal design constraints that align with the user's intent. It also includes a set of example crystals that share the same Wyckoff prototype and anonymized formula prototype.

### 3.4 GENERATION STAGE WITH CRYSTAL DESIGN CONSTRAINTS CHECK

We design a Constraint-based Crystal Generation Agent to learn atomic distribution patterns from a small set of crystal structure examples and generate crystal structures that adhere to predefined design constraints, such as chemical formula and space group. Given a predefined system prompt $p_{sys\_gen}$ for guiding the LLM in few-shot learning, along with a set of example crystals, each defined by their properties and corresponding structures $[(\mathcal{P}_1, \mathcal{M}_1), (\mathcal{P}_2, \mathcal{M}_2), \cdots]$, and the target crystal properties $\mathcal{P}'$, that satisfy the design constraints, the task of the Constraint-based Crystal Generation Agent can be formalized as follows:

$$\mathcal{M}' = \text{LLM}(p_{sys\_gen}, [(\mathcal{P}_1, \mathcal{M}_1), (\mathcal{P}_2, \mathcal{M}_2), \cdots], \mathcal{P}'), \tag{3}$$

$$\text{Check}(\mathcal{M}') = \begin{cases} \text{Valid} & \text{Check}_{c.}(\mathcal{M}', \mathcal{P}') \wedge \text{Check}_{s.}(\mathcal{M}', [\mathcal{M}_1, \cdots]), \\ \text{Invalid} & \text{otherwise.} \end{cases} \tag{4}$$

In this context, $\text{Check}_{c.}(\mathcal{M}', \mathcal{P}')$ represents the check to verify whether the generated crystal structure $\mathcal{M}'$ satisfies the crystal design constraints, while $\text{Check}_{s.}(\mathcal{M}', [\mathcal{M}_1, \cdots])$ represents the check to evaluate whether the generated crystal structure $\mathcal{M}'$ exhibits structural similarity to the example crystal structures. The constraint check $\text{Check}_{c.}$ includes a chemical composition constraint check and a space group constraint check. The similarity check $\text{Check}_{s.}$ comprises a Wyckoff prototype check and an atomic position check, ensuring that the generated structure matches the example Wyckoff prototypes and that atomic coordinates align with the coordinate pattern of the corresponding Wyckoff positions. If the generated structure passes the check, it proceeds to the next stage; otherwise, it is regenerated until valid or the iteration limit is reached.

### 3.5 OPTIMIZATION STAGE FOR OPTIMAL GENERATED STRUCTURE SELECTION

Since the crystal structures generated by the LLM are not guaranteed to lie at their local energy minima, an optimization stage is required to perform structural relaxation and obtain lower-energy structures. To this end, each generated candidate is relaxed using M3GNet (Chen & Ong, 2022), which serves here as an external tool to optimize atomic positions and lattice parameters. The relaxation process minimizes residual atomic forces and lowers the predicted energy, leading the structures to converge toward local minima on the M3GNet potential-energy surface.

After relaxation, a set of candidate structures $\{\tilde{\mathcal{M}}'_1, \tilde{\mathcal{M}}'_2, \ldots, \tilde{\mathcal{M}}'_k\}$ is obtained, each associated with a predicted total energy value $E_{\text{total}}(\tilde{\mathcal{M}}'_i)$. We then compute the corresponding energy per atom as:

$$\bar{E}(\tilde{\mathcal{M}}'_i) = \frac{E_{\text{total}}(\tilde{\mathcal{M}}'_i)}{N_{\text{atom}}}, \tag{5}$$

where $N_{\text{atom}}$ is the number of atoms in the unit cell. Since all candidates are generated under the same completed chemical formula constraint obtained from the retrieval stage, they share the same chemical composition. Therefore, comparing their energy per atom is equivalent to comparing their formation energies up to a constant shift. The final selection is based on the energy per atom:

$$\mathcal{M}^* = \arg \min_{\tilde{\mathcal{M}}'_i \in \{\tilde{\mathcal{M}}'_1, \ldots, \tilde{\mathcal{M}}'_k\}} \bar{E}(\tilde{\mathcal{M}}'_i), \tag{6}$$

where $\mathcal{M}^*$ denotes the selected structure and $\bar{E}(\tilde{\mathcal{M}}'_i)$ is the predicted energy per atom of the relaxed candidate. By choosing the configuration with the lowest energy per atom, the framework produces a final output that is consistent with the design constraints and corresponds to a locally relaxed, low-energy structure.

| | Validity Rate | | Coverage | | Distribution | |
|---|---|---|---|---|---|---|
| | Structural | Composition | Recall | Precision | Density | Element |
| CDVAE | 100.00% | 86.70% | 99.15% | 99.49% | 0.688 | 0.278 |
| DiffCSP | 100.00% | 83.25% | 99.71% | 99.76% | 0.350 | 0.125 |
| FlowMM | 96.85% | 83.19% | 99.49% | 99.58% | 0.239 | 0.083 |
| CrystalTextLLM | 99.60% | 95.40% | 85.80% | 98.90% | 0.810 | 0.440 |
| MatExpert | 99.80% | 96.10% | 98.60% | 99.10% | 0.180 | 0.040 |
| FLowLLM | 99.94% | 90.84% | 96.95% | 99.82% | 1.140 | 0.150 |
| Ours | 99.77% | 97.42% | 98.93% | 99.68% | 0.339 | 0.141 |

Table 1: Crystal Structure Generation.

# 4 EXPERIMENTS

## 4.1 DATASETS

The MP-20 dataset (Jain et al., 2013) is derived from the Materials Project and contains approximately 45,000 crystalline materials with up to 20 atoms per unit cell. It covers a broad chemical space with nearly 90 elements represented and includes both experimentally reported and theoretically computed compounds. The MPTS-52 dataset (Jiao et al., 2023) extends the difficulty by including crystals with up to 52 atoms per unit cell, comprising around 40,000 structures also sourced from the Materials Project. The Challenge Set (Antunes et al., 2024) consists of 70 crystalline compounds, 58 structures are sourced from recent literature and are guaranteed to be absent from the training data, while the remaining 12 are included from the training corpus. In this work, we evaluate the model exclusively on the 58 compounds from recent literature.

## 4.2 CRYSTAL STRUCTURE GENERATION

**Setup.** We first evaluate the performance of our proposed agent on the CSG task. We conduct experiments on the MP-20 dataset, following the data split used in previous works (Xie et al., 2022). Specifically, we use the training set of the MP-20 dataset as the retrieval database for our method, while the test set serves as the evaluation benchmark. In our experiments, we generate 10,000 candidate crystal structures on the MP-20 dataset. The quality of the generated crystal structures is assessed using three key metrics: Validity Rate, Coverage, and Distribution. The Validity Rate evaluates the structural validity and chemical feasibility of the generated structures based on interatomic distances and charge balance. For the Coverage metric, we utilize Recall and Precision, which are calculated using CrystalNN fingerprints Zimmermann & Jain (2020) and normalized Magpie fingerprints Ward et al. (2016). The Property distribution of the generated structures is assessed using the Wasserstein distance for density and the number of distinct element types in the unit cell. These metrics offer a thorough evaluation of how closely the generated structures align with real-world material properties.

**Baselines.** We primarily consider two categories of methods as baselines. One category includes generative models, such as CDVAE (Xie et al., 2022), DiffCSP (Jiao et al., 2023), and FlowMM (Miller et al., 2024), while the other category consists of crystal generation methods based on LLMs, such as FlowLLM (Sriram et al., 2024), CrystalTextLLM Gruver et al. (2024), and MatExpert (Ding et al., 2025). Further details are provided in Appendix B.

**Results.** Our proposed method demonstrates strong performance across all evaluation metrics, as shown in Table 1. In terms of validity, our approach is comparable to or slightly outperforms the best-performing baselines. Notably, it achieves the best chemical feasibility among all methods, highlighting its ability to generate chemically valid crystal structures. Regarding coverage, our method outperforms other LLM-based models in recall, while showing comparable precision. In terms of distribution, our approach performs on par with existing methods. An important distinction is that our model does not require fine-tuning, which underscores its efficiency and robustness, offering a significant advantage over other methods that rely on extensive fine-tuning to achieve optimal results.

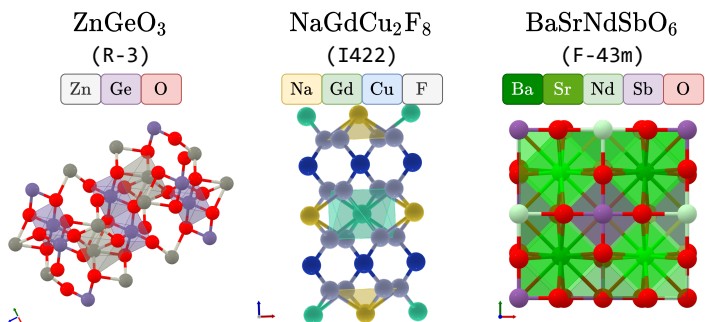

Figure 3: Examples of generated crystal structures in the CSG task.

|  | MP-20 | | MPTS-52 | | Challenge Set | |
|---|---|---|---|---|---|---|
|  | Match Rate | RMSE | Match Rate | RMSE | Match Rate | RMSE |
| CDVAE | 33.9% | 0.105 | 5.3% | 0.211 | - | - |
| DiffCSP | 51.5% | 0.063 | 12.2% | 0.179 | - | - |
| CrystaLLM | 58.7% | 0.041 | 19.2% | 0.111 | 22.4% | 0.090 |
| Mat2Seq | 61.3% | 0.040 | 23.1% | 0.109 | - | - |
| Ours | 64.2% | 0.034 | 31.0% | 0.068 | 32.8% | 0.096 |

Table 2: Crystal Structure Prediction.

## 4.3 CRYSTAL STRUCTURE PREDICTION

**Setup.** In this section, we evaluate the performance of our proposed agent on the CSP task. The evaluation is conducted on the MP-20, MPTS-52, and the Challenge Set. Following previous work (Xie et al., 2022; Jiao et al., 2023; Antunes et al., 2024), we employ two metrics to assess the quality of the generated crystal structures: Match Rate and RMSE. The Match Rate is calculated using pymatgen's StructureMatcher (Ong et al., 2013) to determine the percentage of generated structures that align with the ground truth structures. RMSE is used to measure the structural differences between the generated and ground truth structures.

**Baselines.** We primarily consider two categories of methods as baselines. The first category consists of methods based on diffusion models, such as CDVAE (Xie et al., 2022) and DiffCSP (Jiao et al., 2023), while the second category includes approaches based on GPT-2, such as CrystaLLM (Antunes et al., 2024) and Mat2Seq (Yan et al., 2024). Further details are provided in Appendix B.

**Results.** Our proposed method consistently outperforms existing baselines across all evaluation datasets, as shown in Table 2. It achieves superior Match Rates and lower RMSE values, indicating that the generated crystal structures are closer to the ground truth with minimal structural differences. Notably, our model performs particularly well compared to other methods based on both diffusion models and GPT-2 language models, demonstrating its robustness and ability to handle diverse datasets. Overall, our approach provides a balanced and effective solution for the crystal structure prediction task, delivering improved accuracy and structural consistency without the need for extensive fine-tuning.

## 4.4 CRYSTAL STRUCTURE DESIGN

**Setup.** In this section, we evaluate the ability of our proposed framework to generate crystal structures that satisfy user-specified requirements. We consider three types of tasks: generating crystals with a target band gap value, generating crystals containing specific elements, and generating crystals with a prescribed elemental ratio. Concretely, we require the model to generate crystals with a band gap around 0 eV, crystals containing sulfur (S), and crystals with an elemental ratio of 1:1:3. For each condition, we generate 500 candidate structures and conduct retrieval on the MP-20 dataset. To assess the quality of the generated structures, we use MEGNet (Chen et al., 2019) to estimate their band gaps and employ *pymatgen* to determine whether the generated crystals contain the specified

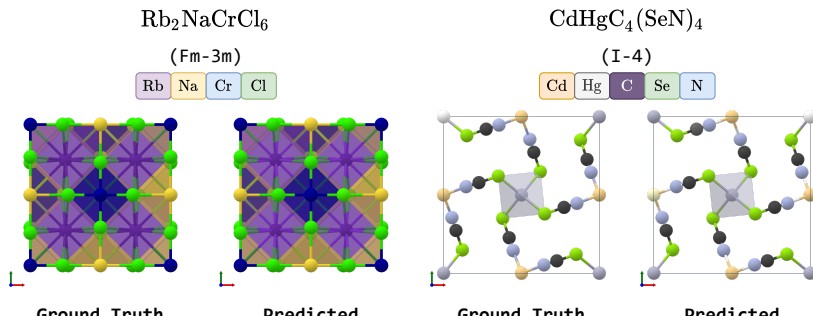

Figure 4: Examples of generated crystal structures in the CSP task.

| Design Objective | Objective Satisfaction | Validity | Uniqueness | Novelty |
|---|---|---|---|---|
| Bandgap around 0 eV | 86.8% | 100.0% | 99.0% | 93.0% |
| Crystal with S element | 96.2% | 100.0% | 76.8% | 92.4% |
| $ABC_3$ type Crystal | 100.0% | 99.4% | 82.5% | 87.9% |

Table 3: Crystal Structure Design.

elements and whether their chemical compositions satisfy the desired elemental ratio. In addition, we further evaluate the validity, uniqueness, and novelty of the generated structures.

**Results.** The performance of our framework in generating crystal structures that satisfy user-specified requirements demonstrates promising results across all three design objectives. The results in Table 3 indicate that our framework is highly effective in generating crystal structures that meet specific physical and chemical constraints, while also exhibiting notable uniqueness and novelty. This highlights the strong capability of our proposed model in fulfilling various requirements for crystal structure design.

## 5 RELATED WORK

### 5.1 GENERATIVE MODELS FOR CRYSTAL GENERATION

Crystal structure generation has been studied with variational autoencoders, diffusion, and flow-based models. CDVAE (Xie et al., 2022) combined a VAE with diffusion-based denoising to generate stable periodic structures. DiffCSP (Jiao et al., 2023) extended this by jointly generating lattice parameters and fractional atomic positions via a periodic equivariant model. FlowMM (Miller et al., 2024) introduced Riemannian flow matching tailored to crystalline symmetries for efficient generation. MatterGen (Zeni et al., 2025) further enhanced diffusion-based models with adapter modules for property-constrained material generation.

### 5.2 AUTOREGRESSIVE LANGUAGE MODELS FOR CRYSTAL GENERATION

Recent advances in LLMs trained on large-scale corpora have demonstrated broad capabilities, motivating researchers to explore the application of LLMs to crystal generation. CrystaLLM (Antunes et al., 2024), trained on CIF files with GPT-2, showed the feasibility of autoregressive modeling. CrystalFormer (Cao et al., 2024) and WyFormer (Kazeev et al., 2025) improved fidelity by incorporating crystallographic priors. CrystalTextLLM (Gruver et al., 2024) fine-tuned LLaMA-2 on XYZ representations, highlighting the potential of natural language guidance in crystal generation. MatExpert (Ding et al., 2025) simulated expert-like structure modification, combining GPT-4 with fine-tuned LLaMA for guided generation.

### 5.3 LLM AGENT FOR CRYSTAL GENERATION

Recent studies have begun to explore the use of LLM-based agents for autonomous materials discovery. AtomAgents (Ghafarollahi & Buehler, 2024) integrates multimodal data and simulations for alloy design. OSDA Agent (Hu et al., 2025a) introduces a generation–evaluation–reflection–refinement workflow for zeolite synthesis. LLMatDesign (Jia et al.,

2024) frames material search as multi-step decision-making with atomic substitutions and deletions. MatLLMSearch (Gan et al., 2025) shows that pre-trained LLMs, combined with evolutionary search, can generate stable crystals without fine-tuning.

# 6 Conclusion

In this work, we introduced CrystalAgent, an LLM-based agent that establishes a fine-tuning free and adaptable framework for crystal generation. The crystal generation process is decomposed into four key stages, namely Extraction, Retrieval, Generation, and Optimization. This design enables systematic and modular handling of diverse crystal generation objectives. Extensive experiments across various crystal generation tasks highlight the flexibility, controllability, and effectiveness of our framework, underscoring the substantial potential of LLM agents in automating the generation of crystal materials and advancing the field of materials discovery.

## Limitations and Future Directions

Our current framework is a deterministic, pre-designed four-stage pipeline and does not perform long-horizon planning or fully autonomous tool selection. We use a simple, fixed similarity-based retrieval strategy with hand-crafted thresholds and do not explore learned or adaptive retrieval policies. We also treat user-specified constraints as hard requirements: if no structurally similar examples are found after constraint completion and retrieval, or if no candidate passes our generation-time checks, the system simply returns no structure. Finally, all validation in this work is purely in silico, without experimental confirmation of synthesizability.

In future work, the framework could be made more agent-like, for example by enabling adaptive tool selection and multi-step refinement based on intermediate feedback. It would also be valuable to incorporate predictors of thermodynamic stability and synthetic accessibility, so that the system reasons about both structural correctness and practical synthesizability.

## Ethics statement

Our study is limited to scientific questions and does not involve human participants, animal experiments, or environmentally sensitive materials. As such, we do not anticipate any ethical concerns or conflicts of interest. We adhere to rigorous standards of scientific integrity and ethics to ensure the reliability, transparency, and validity of our findings.

## Reproducibility Statement

All datasets employed in this work are publicly accessible, with their sources and detailed descriptions provided in Appendix A. Comprehensive implementation details, experimental settings, and evaluation metrics are reported in the Appendix D and Appendix C to support faithful reproduction of our results. For transparency, Appendix G includes the exact prompt templates used by CrystalAgent. Baseline results are taken directly from the respective original papers to ensure a fair and consistent comparison. We believe these resources together provide sufficient guidance for independent verification and extension of our work.

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

## A DETAILS OF DATASETS

The MP-20 dataset (Jain et al., 2013) is derived from the Materials Project and contains approximately 45,000 crystalline materials with up to 20 atoms per unit cell. It covers a broad chemical space with nearly 90 elements represented and includes both experimentally reported and theoretically computed compounds. Due to the moderate cell size, MP-20 has become a widely used benchmark for evaluating models on CSG and CSP. Its diversity of compositions and structures makes it suitable for testing a model's ability to reconstruct, generate, and predict realistic crystals at manageable computational cost.

The MPTS-52 dataset (Jiao et al., 2023) extends the difficulty by including crystals with up to 52 atoms per unit cell, comprising around 40,000 structures also sourced from the Materials Project.

Compared to MP-20, MPTS-52 poses a significantly greater challenge due to the larger and more complex unit cells, which demand stronger generalization and scalability from generative and predictive models. As such, it is frequently used as a stress test for assessing whether methods trained on simpler structures can extrapolate to more realistic, high-complexity crystalline systems.

The Challenge Set (Antunes et al., 2024) curated benchmark designed to rigorously assess the model's ability to generate realistic crystal structures, particularly those not included in existing crystal databases. It consists of 70 crystalline compounds, 58 structures are sourced from recent literature and are guaranteed to be absent from the training data, providing a stringent test of extrapolation, while the remaining 12 are included from the training corpus to assess reproduction of known structures. In this work, we evaluate the model exclusively on the 58 compounds from recent literature, ensuring a rigorous test of the model's ability to generate novel, unseen structures.

## B  DETAILS OF BASELINES

CDVAE (Xie et al., 2022) is a two-stage crystal generation model that combines variational autoencoders and diffusion models. It utilizes the variational autoencoder to generate lattice constants and atom types, while the diffusion model generates atomic coordinates by iteratively denoising the structure.

DiffCSP (Jiao et al., 2023) introduces fractional coordinates in crystal generation tasks and proposes a periodic E(3)-equivariant denoising model that can simultaneously generate lattice constants and atomic coordinates.

FlowMM (Miller et al., 2024) introduces a Riemannian Flow Matching-based approach that improves crystal structure prediction and crystal generation. This method efficiently handles the symmetries inherent in crystals and generates materials with better stability and diversity.

FlowLLM (Sriram et al., 2024) combines the strengths of LLMs and Riemannian flow matching to bridge the gap between discrete and continuous modeling in crystal generation. This model effectively generates novel crystalline materials by integrating both types of modeling.

CrystaLLM (Antunes et al., 2024) is a GPT-2-based model trained on millions of CIF files, focusing on crystal structure generation and prediction tasks.

Mat2Seq (Yan et al., 2024) introduces a method to convert 3D crystal structures into 1D sequences, ensuring that each unique crystal structure is represented as a distinct sequence. Like CrystaLLM, it uses GPT-2 as the backbone architecture and ensures SE(3) and periodic invariance, enabling efficient generation of crystal structures with language models.

CrystalTextLLM (Gruver et al., 2024) leverages a fine-tuned LLaMA-2 model for crystal generation tasks, using the XYZ format of crystal structure for generation. It supports both unconditional and conditional crystal generation in zero-shot scenarios, showcasing the capability to generate diverse crystal structures based on text inputs.

MatExpert (Ding et al., 2025) decomposes the crystal generation process into three stages: retrieval, transition, and generation. Mimicking the expert-driven workflow in materials discovery, it utilizes the transition pathway data generated by GPT-4o to fine-tune LLaMA-2 and LLaMA-3 models, facilitating more efficient material design.

## C  DETAILS OF METRICS

For the CSG task, we adopt three categories of evaluation metrics: validity rate, coverage, and distribution. The validity rate measures both structural validity and chemical feasibility of the generated crystals by checking interatomic distances and charge balance. Coverage is assessed through recall and precision computed with CrystalNN fingerprints (Zimmermann & Jain, 2020) and normalized Magpie fingerprints (Ward et al., 2016). Recall quantifies the fraction of test set structures that are successfully generated, while precision captures the fraction of generated structures that appear in the test set. To further characterize the quality of generation, we analyze property distributions of the generated crystals. Specifically, we compute the Wasserstein distance on two key attributes, the

| Variant | MP-20 | |
|---|---|---|
| | Match Rate | RMSE |
| w/o constraint completion | 42.8% | 0.035 |
| w/o similarity-based retrieval | 57.8% | 0.043 |
| w/o generation-time checks | 64.2% | 0.035 |
| w/o optimization | 62.9% | 0.054 |
| Ours | 64.2% | 0.034 |

Table 4: Ablation study on the CSP task for MP-20.

density and the number of distinct element types in a unit cell, in order to compare the generated data against real-world material statistics.

For the CSP task, we adopt match rate and RMSE. The match rate is computed with pymatgen's StructureMatcher (Ong et al., 2013), which identifies whether a generated structure aligns with the ground-truth crystal. RMSE measures the geometric difference between atomic positions of generated and target structures, thereby quantifying the structural accuracy of prediction results.

For the CSD task, we evaluate the generated structures using MEGNet (Chen et al., 2019) to estimate their band gaps and employ *pymatgen* to verify whether the crystals contain the specified elements and whether their chemical compositions satisfy the required elemental ratios. We further assess validity, uniqueness, and novelty. Uniqueness quantifies how many distinct structures exist within the generated set, and novelty evaluates how many of the generated structures do not appear in the retrieval set.

## D  EXPERIMENT SETTING

In our experiments, we use the GPT-4o-mini-2024-07-18 model as the underlying LLM. All experiments are conducted on a single NVIDIA A40 GPU, primarily used for running M3GNet to relax the crystal structures generated by the LLM. For the CSG task, we employ the train set of the MP-20 dataset as the retrieval pool and generate 10,000 candidate structures for evaluation. For the CSP task, we evaluate on the MP-20, MPTS-52, and Challenge Set datasets. For the CSD task, we generate 500 candidate structures for each task.

## E  ABLATION STUDY

To quantify the contribution of the main components in our four-stage pipeline, we conduct an ablation study on the CSP task using the MP-20 dataset. We follow the standard split used in prior work and use the training split as the retrieval pool while evaluating on the test split.

Table 4 reports the Match Rate and RMSE for several variants of CrystalAgent:

- **w/o constraint completion**: we replace the space-group prediction tool used to complete missing symmetry constraints from the given composition with a simple prior based on the empirical distribution of space groups for anonymized compositions in the retrieval database.
- **w/o similarity-based retrieval**: we remove the retrieval-time similarity constraints based on anonymized oxidation-state formulas, elemental distances, and Wyckoff prototypes when selecting example structures, and instead sample examples that only satisfy the basic composition and space-group constraints.
- **w/o generation-time checks**: we disable the crystal-structure validity checks applied after generation and accept the first structure returned by the LLM for each query.
- **w/o optimization**: we remove the post-generation optimization stage based on M3GNet relaxation and directly use the first valid structure that passes the generation-time checks.

Removing the constraint-completion module (w/o constraint completion) leads to a substantial drop in Match Rate, highlighting the importance of accurate completion of missing symmetry constraints. Turning off the retrieval-time similarity constraints (w/o similarity-based retrieval) also degrades both Match Rate and RMSE, indicating that structurally similar examples are essential for guiding the model toward high-quality, constraint-satisfying structures. Disabling the generation-time checks (w/o generation-time checks) yields performance that is very close to the full model, suggesting that our crystal-structure representation and constraints already lead the model to produce valid structures in most cases. Finally, skipping the optimization stage (w/o optimization) slightly reduces the Match Rate and increases the RMSE, showing that the explicit optimization module is still helpful for refining LLM-generated candidates toward lower-energy structures.

## F    Algorithmic Description of the Retrieval Stage

To complement the description in Section 3.3, we provide a step-by-step algorithmic breakdown of the Constraints-based Similar Structure Retrieval Agent used in the Retrieval stage.

## G    Prompts

To clarify how CrystalAgent interacts with LLMs, we provide the exact prompt templates used in our framework. Since prompt design directly affects the behavior and output quality of LLMs, we make them available here to ensure transparency and reproducibility. The prompts are divided into two categories according to their role in the pipeline.

The first category is devoted to formula generation, where the model is guided to produce chemically valid formulas.

```
You are a materials scientist assistant.

Your task is to generate a chemically reasonable crystal formula
↪  based on a few example compounds. You should consider:
- The oxidation states and charge balance of elements
- Periodic group and chemical similarity between elements
- Structural patterns or common motifs found in the examples

Important:
- Do **not** repeat any of the formulas from the examples.
- Only generate a new, plausible chemical formula that shares
↪  similar chemistry but is distinct from the given ones.

Here are the few-shot examples:
{example}
Please generate a new plausible chemical formula and briefly
↪  explain your reasoning.
```

The second category targets crystal structure generation, where the model receives demonstration examples and constraint specifications and is instructed to produce complete crystal structures in the designated representation format.

```
You are very powerful assistant for various crystal-related tasks
↪  from diverse user inputs.
You can learn the laws of crystal structure from a small number of
↪  examples input by the user and generate a crystal structure
↪  that meets the user's requirements:

Here are several similar crystal property descriptions and
↪  corresponding structure description texts:
{fewshot_example}
```

---

**Algorithm 1** Constraints-based Similar Structure Retrieval Agent

---

**Require:** User constraints $(c_f, c_{sg}, c_p, c_{af}, c_e)$, structure database $\mathcal{D}$, LLM, space-group prediction tool SGP, number of examples $K$, maximum formula-generation attempts $T_{\max}$
**Ensure:** Completed constraints $(\hat{c}_f, \hat{c}_{sg}, \hat{c}_p, \hat{c}_{af}, \hat{c}_e)$ and example set $\mathcal{E}$
1: Initialize $(\hat{c}_f, \hat{c}_{sg}, \hat{c}_p, \hat{c}_{af}, \hat{c}_e) \leftarrow (c_f, c_{sg}, c_p, c_{af}, c_e)$
2: **if** $\hat{c}_f \neq \emptyset$ **and** $\hat{c}_{sg} \neq \emptyset$ **then**
3:     Filter $\mathcal{D}$ using $(\hat{c}_f, \hat{c}_{sg}, \hat{c}_p, \hat{c}_{af}, \hat{c}_e)$ to obtain $\mathcal{R}_{\text{final}}$
4: **else if** $\hat{c}_f \neq \emptyset$ **and** $\hat{c}_{sg} = \emptyset$ **then**
5:     Use SGP conditioned on $\hat{c}_f$ to predict a space group $\hat{c}_{sg}$
6:     Filter $\mathcal{D}$ using $(\hat{c}_f, \hat{c}_{sg}, \hat{c}_p, \hat{c}_{af}, \hat{c}_e)$ to obtain $\mathcal{R}_{\text{final}}$
7: **else if** $\hat{c}_f = \emptyset$ **then**
8:     **if** $\hat{c}_{sg} = \emptyset$ **then**
9:         Retrieve an initial candidate pool $\mathcal{R}_0 \subset \mathcal{D}$ that matches any available $(\hat{c}_{af}, \hat{c}_e, \hat{c}_p)$
10:     **else**
11:         Retrieve an initial candidate pool $\mathcal{R}_0 \subset \mathcal{D}$ that matches $\hat{c}_{sg}$ and any available $(\hat{c}_{af}, \hat{c}_e, \hat{c}_p)$
12:     **end if**
13:     Group $\mathcal{R}_0$ by anonymized formula and space group
14:     Select a group $\mathcal{G}$ that is consistent with any available $(\hat{c}_{af}, \hat{c}_e, \hat{c}_p)$
15:     **if** $\hat{c}_{sg} = \emptyset$ **then**
16:         Set $\hat{c}_{sg}$ to the space group associated with $\mathcal{G}$
17:     **end if**
18:     $t \leftarrow 0$
19:     **while** $t < T_{\max}$ **do**
20:         Use the LLM to infer a plausible chemical formula $\hat{c}_f$ from the examples in $\mathcal{G}$
21:         Check whether $\hat{c}_f$ is electroneutral
22:         Check whether the anonymized formula of $\hat{c}_f$ matches that of $\mathcal{G}$
23:         **if** both checks pass **then**
24:             **break**
25:         **else**
26:             $t \leftarrow t + 1$
27:         **end if**
28:     **end while**
29:     **if** $t = T_{\max}$ **then**
30:         Report failure and terminate retrieval
31:     **else**
32:         Filter $\mathcal{D}$ using $(\hat{c}_f, \hat{c}_{sg}, \hat{c}_p, \hat{c}_{af}, \hat{c}_e)$ to obtain $\mathcal{R}_{\text{final}}$
33:     **end if**
34: **end if**
35: Group $\mathcal{R}_{\text{final}}$ by Wyckoff prototype
36: Select a Wyckoff prototype $\pi$ with at least $K$ members
37: Sample up to $K$ examples $\mathcal{E} \subset \mathcal{R}_{\text{final}}$ that share prototype $\pi$
38: **return** $(\hat{c}_f, \hat{c}_{sg}, \hat{c}_p, \hat{c}_{af}, \hat{c}_e)$ and example set $\mathcal{E}$

---

```
Please generate the corresponding crystal structure description
↪  text for the following crystal properties based on the small
↪  number of examples provided. You should produce output in
↪  exactly the same format as the example, without including any
↪  extraneous characters:
{crys_attr}
```

## H  THE USE OF LARGE LANGUAGE MODELS

For the purpose of improving readability and presentation, we employed the LLM solely as a tool for linguistic refinement. Its use was limited to tasks such as grammar correction and proofreading, comparable to the functions of conventional editing software and dictionaries. Importantly, the

LLM did not generate or influence the scientific content of this work, and its application aligns with common practices in academic manuscript preparation.

