# OpenReview forum: "CrystalAgent: Towards Autonomous Crystal Generation via Agentic Reasoning"
_ICLR.cc/2026/Conference — Submitted to ICLR 2026_

### Official Review · Reviewer_Fxuo · 2025-10-24

**Soundness:** 3
**Presentation:** 4
**Contribution:** 3
**Rating:** 8
**Confidence:** 3

**Summary:**

The work proposes an agentic framework (CrystalAgent) for crystal generation with the main claim that CrystalAgent eliminates the need for additional training and provides more flexibility in comparison with the current solutions in diverse crystal discovery setups. Based on user query consisting of design constraints, CrystalAgent retrieves few-shot examples best matching the query based on element proportions (anonymized formula) and structural features (Wyckoff prototype) represented in symmetry-based encoding and used for conditional generation by LLM. Framework was validated in one unconditional and two conditional tasks including i) unconditional crystal structure generation (CSG), ii) structure prediction (CSP), and iii) structure design (CSD).

**Strengths:**

1. Genuinely relevant challenge of flexible crystal design.
2. High clarity and design of the paper.
3. Accessible explanation of mathematical concepts used in the work.
4. Broad framework validation in three setups (CSG, CSP, and CSD).

**Weaknesses:**

1. The rationale behind the use of LLM for crystal generation is not well posed in the paper, where plenty of models exist that work on graph structures, intrinsically respect symmetries, and seem to be more natural to use while LLM is often the best choice for data natively represented by text. Moreover, it would be necessary to probe other LLMs besides gpt-4o on these tasks for comprehensive comparison since performance on the specific tasks often depends strongly on the model.
2. Some of the claims (e.g., "dependence on extensive training data restricts generalization" in line 62) are either poorly formulated or incorrect since contradict with well-established observations, which should be checked more carefully. On the contrary, informing the model with few-shot examples sampled from MP based on the specified constraints is more biased and can restrict the generalization. Authors should pay more attention to retrieval stage e.g., ways of informing the model with examples, optimal thresholds for examples retrieval etc.
3. Benchmarking against more recent SOTA models should be performed (DiffCSP++, con-CDVAE, WyckoffDiff, MatterGen, SymmCD), where CSD task also lacks comparison with existing solutions or/and some baselines.
4. Thermodynamic stability of generated materials should be evaluated (formation energy, energy above the convex hull).

**Questions:**

1. What will the framework do if so happens there are no structures matching the constraints close enough? What happens if the specified set of constraints can't be satisfied theoretically? Such cases should be studied in more details to find edge cases characterizing framework limitations.
2. Are fractional coordinates optimal (many cells can describe the same crystal, orientation/permutation variance for F, not ensures fine interatomic distances, nearly singular L are easy to generate but hard to relax etc.)?

---

> ### Author Response · Authors · 2025-12-02
> **Response to Reviewer Fxuo (1/5)**
>
> We are truly grateful for your positive feedback! We also sincerely thank you for your valuable suggestions and would like to respond to your concerns one by one in the following.
>
> **Weakness 1 -- The rationale behind the use of LLM for crystal generation is not well posed in the paper, where plenty of models exist that work on graph structures, intrinsically respect symmetries, and seem to be more natural to use while LLM is often the best choice for data natively represented by text. Moreover, it would be necessary to probe other LLMs besides gpt-4o on these tasks for comprehensive comparison since performance on the specific tasks often depends strongly on the model.**
>
> **Response:** We thank the reviewer for raising this important point. Our goal in this work is not to claim that LLMs are inherently superior to graph-based or symmetry-aware generative models for crystal data, but to explore how a large language model can be used as the core of a flexible crystal structure generation agent when combined with explicit constraints and external tools. In particular, CrystalAgent is designed to (1) accept high-level and mixed-format user specifications (natural language, formulas, partial constraints), (2) automatically complete missing design constraints, and (3) orchestrate retrieval, validation tools, and optimization in a single workflow. This type of interactive, constraint-driven design process, which tightly integrates querying, reasoning, and tool use, is less natural to implement with purely graph-based or diffusion-based models that are typically trained for fixed input–output formats.
>
> At the same time, we fully agree that graph-based and symmetry-aware models are very natural for crystal data, and we do not position CrystalAgent as a replacement for them. Several such models are included in our baselines, and our results show that the proposed LLM-based framework can be competitive with these specialized architectures on CSP/CSG benchmarks while offering a different set of advantages, such as flexible constraint handling and easy integration with external optimizers.
>
> Regarding the choice of LLM backbone, the CrystalAgent framework itself is model-agnostic: the constraint completion, retrieval, generation, and optimization stages do not rely on any GPT-4o-specific feature. We chose GPT-4o as a strong, widely used backbone to probe the potential of this agent-style framework under a high-capacity model. We agree that a systematic comparison across different LLMs would provide a more complete picture and that performance can depend on the underlying model, so we have added experiments using LLaMA-3 70B within our method. The comparative results are reported in the table below.
>
> |                   |   MP-20    |       |  MPTS-52   |       |
> | :---------------: | :--------: | :---: | :--------: | :---: |
> |                   | Match Rate | RMSE  | Match Rate | RMSE  |
> |       CDVAE       |   33.9%    | 0.105 |    5.3%    | 0.211 |
> |      DiffCSP      |   51.5%    | 0.065 |   12.2%    | 0.179 |
> |     CrystaLLM     |   58.7%    | 0.041 |   19.2%    | 0.111 |
> |      Mat2Seq      |   61.3%    | 0.040 |   23.1%    | 0.109 |
> | Ours(Llama3-70B)  |   65.1%    | 0.034 |   32.0%    | 0.067 |
> | Ours(GPT-4o-mini) |   64.2%    | 0.034 |   31.0%    | 0.068 |
>
> Replacing GPT-4o-mini with LLaMA-3 70B leads to only minor fluctuations in performance, and our method continues to outperform the baselines by a similar margin. This suggests that the gains primarily stem from the proposed agentic pipeline rather than from a particular choice of backbone.
> More specifically, because the current Ollama implementation of LLaMA-3 does not support the tool-calling interface we rely on to strictly constrain output formats, we only swap the Generation component to LLaMA-3 70B. The structured-output components of our pipeline (for example, crystal-constraint extraction) still use GPT-4o-mini with tool calls. This setup isolates the effect of changing the generative backbone while keeping the downstream structured reasoning unchanged, and the stability of results across GPT-4o-mini and LLaMA-3 70B further supports that our improvements are not an artifact of a stronger single model.

---

> > ### Author Response · Authors · 2025-12-02
> > **Response to Reviewer Fxuo (2/5)**
> >
> > **Weakness 2 -- Some of the claims (e.g., "dependence on extensive training data restricts generalization" in line 62) are either poorly formulated or incorrect since contradict with well-established observations, which should be checked more carefully. On the contrary, informing the model with few-shot examples sampled from MP based on the specified constraints is more biased and can restrict the generalization. Authors should pay more attention to retrieval stage e.g., ways of informing the model with examples, optimal thresholds for examples retrieval etc.**
> >
> > **Response:** We thank the reviewer for this insightful comment and for pointing out the ambiguity in our wording.
> >
> > Our intention in the sentence “dependence on extensive training data restricts generalization” was not to claim that large training datasets inherently harm generalization. The point we meant to make is more specific: many existing crystal generation models are trained in a task-specific, fully supervised fashion on a relatively narrow distribution. In such settings, extending to new types of crystal structure types often requires additional retraining or re-engineering of the model and its training data, which limits practical flexibility. In the revised manuscript, we have rephrased this statement to avoid suggesting that more data is intrinsically bad, and instead emphasize that our framework aims to reduce the need for task-specific retraining by using a general-purpose LLM wrapped in a constraint-aware workflow.
> >
> > We agree that the retrieval stage is crucial and can influence the conditional distribution the model sees. Conditioning the LLM on few-shot examples from the retrieval database does bias the generation toward regions of structure space represented in the database, but this is precisely the design goal: given explicit design constraints, we aim to generate structures that are consistent with those constraints and close to known, physically plausible crystals, rather than exploring arbitrarily far out-of-distribution configurations. To mitigate pathological bias, our Retriever does not use arbitrary examples; it selects structurally similar crystals based on anonymized chemical formulas and Wyckoff prototypes, together with the user-specified constraints. As shown in our ablation study, removing the similarity-based retrieval step leads to a clear degradation in both Match Rate and RMSE, which indicates that, in practice, the retrieved examples help rather than hurt generalization under the evaluated CSP tasks.
> >
> > |                                |    MP20    |       |
> > | :----------------------------: | :--------: | :---: |
> > |                                | Match Rate | RMSE  |
> > | w/o similarity-based retrieval |   57.8%    | 0.043 |
> > |              Ours              |   64.2%    | 0.034 |
> >
> > We agree that the design of the retrieval policy (how examples are selected, how many to use, and what similarity thresholds to apply) is an important component. In our current implementation, we use a simple, fixed strategy: similarity-based filtering followed by a small number of examples. We did not explore more sophisticated retrieval schemes or extensive threshold tuning in this work. We now explicitly acknowledge this as a limitation and a promising direction for future improvement, where more principled or adaptive retrieval strategies could further strengthen or regularize the behavior of CrystalAgent.

---

> > > ### Author Response · Authors · 2025-12-02
> > > **Response to Reviewer Fxuo (3/5)**
> > >
> > > **Weakness 3 -- Benchmarking against more recent SOTA models should be performed (DiffCSP++, con-CDVAE, WyckoffDiff, MatterGen, SymmCD), where CSD task also lacks comparison with existing solutions or/and some baselines.**
> > >
> > > **Response:** We thank the reviewer for pointing out these recent models. Their problem settings are, however, not fully aligned with the evaluations settings in our work.
> > >
> > > con-CDVAE mainly focuses on incorporating conditional constraints into the crystal generation process and reports the success rate of satisfying those conditions, rather than performance on standard CSG or CSP benchmarks that are directly comparable to ours. WyckoffDiff is designed to generate crystal prototypes, that is, the combination of space group, elements, and Wyckoff positions, instead of full crystal structures with atomic coordinates. As a result, it cannot be directly evaluated under CSP and CSG benchmarks, which require complete crystal structures. MatterGen uses a different evaluation setup for CSG from ours. Reproducing its exact protocol and aligning it with our benchmarks would require a dedicated implementation effort, which we plan to consider in future work.
> > >
> > > For DiffCSP++ and SymmCD, whose settings are closer to our CSG task, we extract the reported results from their original papers and list them in a comparison table below. Including DiffCSP++ and SymmCD in our CSG comparisons does not change our main conclusions: their performance falls within the range of baselines we already consider, and CrystalAgent continues to match or improve upon the existing baselines under our evaluation protocol.
> > >
> > > |                | Validity Rate↑ |             | Coverage↑ |           | Distribution↓ |         |
> > > | :------------: | :------------: | :---------: | :-------: | :-------: | :-----------: | :-----: |
> > > |                |   Structural   | Composition |  Recall   | Precision |    Density    | Element |
> > > |     CDVAE      |    100.00%     |   86.70%    |  99.15%   |  99.49%   |     0.688     |  0.278  |
> > > |    DiffCSP     |    100.00%     |   83.25%    |  99.71%   |  99.76%   |     0.350     |  0.125  |
> > > |     FlowMM     |     96.85%     |   83.19%    |  99.49%   |  99.58%   |     0.239     |  0.083  |
> > > | CrystalTextLLM |     99.60%     |   95.40%    |  85.80%   |  98.90%   |     0.810     |  0.440  |
> > > |   MatExpert    |     99.80%     |   96.10%    |  98.60%   |  99.10%   |     0.180     |  0.040  |
> > > |    FLowLLM     |     99.94%     |   90.84%    |  96.95%   |  99.82%   |     1.140     |  0.150  |
> > > |   DiffCSP++    |     99.94%     |   85.12%    |  99.73%   |  99.59%   |     0.235     |  0.375  |
> > > |     SymmCD     |     90.34%     |   85.81%    |  99.58%   |  97.76%   |     0.230     |  0.400  |
> > > |      Ours      |     99.77%     |   97.42%    |  98.93%   |  99.68%   |     0.339     |  0.141  |
> > >
> > > For the CSD task, our goal is mainly to demonstrate that CrystalAgent can adapt to different crystal design objectives and interaction modes. To the best of our knowledge, there is no unified formulation of this CSD setting that can be directly applied to existing baselines without substantial redesign, so we treat CSD as a case study of the flexibility of our framework rather than as a benchmark with standard off-the-shelf baselines.

---

> > > > ### Author Response · Authors · 2025-12-02
> > > > **Response to Reviewer Fxuo (4/5)**
> > > >
> > > > **Weakness 4 -- Thermodynamic stability of generated materials should be evaluated (formation energy, energy above the convex hull).**
> > > >
> > > > **Response:** We thank the reviewer for this valuable suggestion. In the present work, we focus primarily on structural accuracy metrics to evaluate the similarity between the generated structures and the distribution of ground-truth structures. We fully agree that stability is crucial for practical materials discovery, and incorporating formation-energy and convex-hull–based evaluations into CrystalAgent is an important direction that we plan to pursue in future work.
> > > >
> > > > **Question 1 -- What will the framework do if so happens there are no structures matching the constraints close enough? What happens if the specified set of constraints can't be satisfied theoretically? Such cases should be studied in more details to find edge cases characterizing framework limitations.**
> > > >
> > > > **Response:** We thank the reviewer for raising this important point. In our current implementation, satisfying the specified constraints is treated as a hard requirement rather than a soft preference. **If the retrieval database does not contain any structurally similar examples** after constraint completion and retrieval, **the framework terminates** without entering the generation stage and returns no candidate crystal structures. Likewise, during generation we limit the number of regeneration attempts and enforce strict post-generation checks; if no candidate passes these checks within the allowed attempts, the system aborts generation and reports failure instead of returning a structure that violates the constraints.
> > > >
> > > > **When the specified set of constraints is theoretically unsatisfiable, the framework typically fails to find any valid examples in the retrieval database.** In such cases, **it simply does not generate any structure**. We have clarified this behavior in the revised manuscript by explicitly discussing it in the limitation section.

---

> > > > > ### Author Response · Authors · 2025-12-02
> > > > > **Response to Reviewer Fxuo (5/5)**
> > > > >
> > > > > **Question 2 -- Are fractional coordinates optimal (many cells can describe the same crystal, orientation/permutation variance for F, not ensures fine interatomic distances, nearly singular L are easy to generate but hard to relax etc.)?**
> > > > >
> > > > > **Response:** We thank the reviewer for raising this point and agree that fractional coordinates are not an “optimal” representation in a strict sense. We chose them mainly for practicality and compatibility with Wyckoff-based constraints, rather than claiming they are theoretically ideal.
> > > > >
> > > > > First, we use fractional coordinates because they make it convenient to work with **Wyckoff positions**. In our framework, each atom is associated with a Wyckoff position, and it is straightforward to enforce and check this correspondence when positions are expressed in fractional coordinates within a given space group.
> > > > >
> > > > > Regarding the **choice of cell**, we use the **conventional cell** as provided by pymatgen, which tends to preserve the crystal symmetry as much as possible under a standardized convention. This reduces, though does not completely remove, the ambiguity that many equivalent cells can describe the same infinite crystal.
> > > > >
> > > > > For the **permutation variance of (F)** (the list of fractional coordinates), we impose a deterministic ordering: atoms are sorted by Wyckoff position, so that structures with the same Wyckoff pattern are represented in a consistent way. This does not eliminate all possible permutations, but it stabilizes the representation with respect to the Wyckoff-position layout, which is the main object we model.
> > > > >
> > > > > We agree that we do not explicitly model **orientation invariance**. Instead, we use the conventional cell returned by pymatgen to standardize the lattice orientation, and we compute metrics such as Match Rate and RMSE on these standardized structures. In other words, we rely on this conventionalization step during evaluation rather than designing a fully rotation-invariant representation.
> > > > >
> > > > > For the **lattice (L)**, we do not generate an arbitrary ($3 \times 3$) matrix. Instead, we parameterize the lattice by the **3 lengths and 3 angles** of the conventional cell, and these parameters are constrained by the space group. This makes nearly singular or highly pathological cells much less likely in practice. Any remaining geometric issues are further mitigated by the final relaxation step with M3GNet.
> > > > >
> > > > > We acknowledge that more invariant and well-conditioned representations (for example, canonicalized cells or graph-based encodings that factor out symmetry, orientation, and permutations) could further reduce these issues. Exploring such unified encodings within the CrystalAgent framework is an interesting direction for future work, but in this paper we adopt the fractional-coordinate representation because it aligns well with our Wyckoff-based constraint design.

---

### Official Review · Reviewer_3mDi · 2025-10-31

**Soundness:** 1
**Presentation:** 2
**Contribution:** 1
**Rating:** 2
**Confidence:** 4

**Summary:**

The paper presents the framework CrystalAgent which uses large language model agents to generate crystalline materials autonomously. The workflow comprises four stages: extract design constraints from user input, retrieve relevant few-shot examples from a crystal structure database, generate candidate crystal structures with an LLM trained via in-context examples, and finally optimize those structures using physics or energy tools to select the most stable output. Experiments across multiple crystal generation tasks are presented to demonstrate flexibility and adaptability of the approach to diverse crystal discovery scenarios.

**Strengths:**

1. The paper introduces CrystalAgent, an LLM-based framework for autonomous crystal generation, which is a conceptually interesting and timely idea for materials discovery.
2. The motivation to bridge data-driven CSP/CSD and flexible LLM reasoning is well framed.
3. The manuscript is generally well organized and written clearly.

**Weaknesses:**

1. No code is provided despite the reproducibility statement.
2. Missing discussion of limitations and future directions.
3. Retriever design lacks clarity and has no ablation support.
4. Claims about bridging data scarcity are unsubstantiated.
5. The evaluation set appears too small and not well justified.
6. Baseline comparisons are inconsistent and omit important recent methods.
7. Quantitative results lack error bars or deviations.
8. Validation is purely in silico without supporting artifacts or data release.

**Questions:**

1. _“...we propose CrystalAgent, an LLM-based agent that eliminates the need for additional training and adapts flexibly to diverse crystal discovery scenarios.”_

How is this demonstrated? I do not find specific examples of that in the paper.

2. The title “towards” assumes the proposed framework is not definitive but rather an important step in the direction of autonomous agent-based crystal generation. In this context, it is crucial to have a dedicated section on limitations of the current approach and prospects of the agentic frameworks. Both are missing at the moment. Could the authors address this?

3. _“In the Retrieval stage, based on the extracted constraints, the system automatically selects few-shot examples from the database to inform subsequent processes.”_

Retriever design would benefit a lot from a visual representation of its components and key steps or an algorithm highlighting the logical steps. Are there any ablation studies supporting the described design? Could the authors include a schematic or algorithmic breakdown of the Retriever stage?

4. Up until Section 4, the authors refer to the database as the data source used by the proposed framework. However, even from Section 4.1 it is not entirely clear what exactly is that database. Could the authors clarify?

5. _“The Challenge Set (Antunes et al., 2024) consists of 70 crystalline compounds… we evaluate on the 58 compounds from recent literature.”_

Was the framework validated using these 58 compounds? How is it guaranteed they are absent from the training data? Which training data?

6. The test set of 58 compounds seems unreasonably small. Could the authors explain this design choice in detail, including aspects of generalizability, representativeness, and bias in evaluation metrics?

7. Why is the set of baselines different across Tables 1 and 2?

8. Why don’t the authors present error bars or deviations for any evaluations?

9. In my view, Table 1 does not present convincing evidence of the effectiveness of CrystalAgent in the CSG task. Structural Validity Rate and Coverage metrics are not informative since all methods score similarly. For the other metrics, CrystalAgent is somewhat better than CrystalTextLLM, FLowLLM, and CDVAE in Composition Validity Rate, but significantly worse otherwise. The authors advocate that another advantage of CrystalAgent is that it does not require fine-tuning, but I don’t see this as a serious argument in this context. Can the authors provide additional experimental results to support their claims?

10. Table 2 is suggesting that CrystalAgent is superior to other methods, however, there are multiple questions to this comparative analysis. Why is the Challenge Set evaluation limited to a single baseline, i.e. CrystaLLM? This appears as a major limitation. Also, to my knowledge, there are at least two versions of CrystaLLM – which one was used?

11. Table 3 shows promising results, but I find no data assets, repositories, or artifacts supporting this. Can the authors provide them?

12. More generally, do the authors intend to release code, data assets, or artifacts to support reproducibility? Why aren’t those included in the submission?

13. Were only in silico models and tools used for validation?

14. The paper omits some recent and relevant works, e.g. SymmCD, DiffCSP++, deCIFer. Could the authors discuss how CrystalAgent compares to these methods conceptually and experimentally?

15. The authors claim that CrystalAgent bridges the gap caused by data scarcity, but the framework relies on a single database with the same limitations. How is this gap addressed in practice?

16. Can properties of experimental synthesis or synthetic accessibility by itself be important constraints for the CSP and CSD tasks? Why aren’t these considered or discussed?

17. The prompts in Appendix appear quite simple. Did the authors use prompt optimization tools or techniques? Please describe your approach to prompt and context engineering.

---

> ### Author Response · Authors · 2025-12-02
> **Response to Reviewer 3mDi (1/7)**
>
> We would like to address the questions you pointed out one by one below.
>
> **Question 1 -- “...we propose CrystalAgent, an LLM-based agent that eliminates the need for additional training and adapts flexibly to diverse crystal discovery scenarios.”
> How is this demonstrated? I do not find specific examples of that in the paper.**
>
> **Response:** We address the reviewer’s concern as follows. The claim that **CrystalAgent** does not require additional training refers to the fact that our system **operates entirely through inference-time reasoning with a pretrained LLM and a set of external tools**. CrystalAgent is not fine-tuned, adapted, or optimized on any task-specific dataset; all components remain fixed during evaluation. This is why we describe the framework as eliminating the need for additional training.
>
> **Regarding the ability to flexibly adapt to diverse crystal discovery scenarios, this is demonstrated through the three distinct tasks evaluated in the paper**: crystal structure generation (CSG), crystal structure prediction (CSP), and crystal structure design (CSD). These tasks differ substantially in their objectives, inputs, and constraints, yet CrystalAgent employs the same agentic workflow to address all of them without any task-specific retraining or reconfiguration. The successful execution of these three settings illustrates the framework’s ability to operate across multiple crystal discovery scenarios.
>
> **Question 2 -- The title “towards” assumes the proposed framework is not definitive but rather an important step in the direction of autonomous agent-based crystal generation. In this context, it is crucial to have a dedicated section on limitations of the current approach and prospects of the agentic frameworks. Both are missing at the moment. Could the authors address this?**
>
> **Response:** We thank the reviewer for this thoughtful comment. Our intention with the word “towards” in the title is precisely to signal that the proposed framework is an initial step toward agent-style, tool-augmented LLM workflows for crystal structure generation, rather than a definitive or fully autonomous agent system. In the revised version, we have added an explicit discussion of the limitations of our current approach, including its largely deterministic, pre-designed workflow and its reliance on hand-crafted constraint schemas and retrieval heuristics. We have also added a brief subsection on future prospects of agent-based frameworks, outlining how our current system could be extended toward more adaptive agents, for example by allowing the model to make more autonomous tool-selection and planning decisions. This clarification make the “towards” in the title and the relationship to broader agentic frameworks more explicit.

---

> > ### Author Response · Authors · 2025-12-02
> > **Response to Reviewer 3mDi (2/7)**
> >
> > **Question 3 -- “In the Retrieval stage, based on the extracted constraints, the system automatically selects few-shot examples from the database to inform subsequent processes.”
> > Retriever design would benefit a lot from a visual representation of its components and key steps or an algorithm highlighting the logical steps. Are there any ablation studies supporting the described design? Could the authors include a schematic or algorithmic breakdown of the Retriever stage?**
> >
> > **Response:** We appreciate the reviewer’s suggestion and have both clarified the Retriever design and added supporting experiments.
> >
> > **The Retriever is relatively complex because it adapts its behavior to the completeness of the extracted constraints and uses retrieved examples both to complete missing constraints and to select structurally similar demonstrations.** For the CSP task on MP20, for example, we first use a space-group prediction tool to complete missing symmetry information from the given composition. We then perform similarity-based retrieval over the database using the completed chemical formula and space group to obtain structurally similar examples.
> >
> > To quantify the contribution of these components, we conducted an ablation study on the MP20 dataset for CSP, summarized below:
> >
> > |                                |    MP20    |       |
> > | :----------------------------: | :--------: | :---: |
> > |                                | Match Rate | RMSE  |
> > |   w/o constraint completion    |   42.8%    | 0.035 |
> > | w/o similarity-based retrieval |   57.8%    | 0.043 |
> > |              Ours              |   64.2%    | 0.034 |
> >
> > For **w/o constraint completion**, we replace the space-group prediction tool used to complete missing symmetry constraints from the given composition with a simple prior based on the distribution of space groups for anonymized compositions in existing retrieval databases. This leads to a substantial drop in Match Rate, highlighting the importance of an accurate constraint-completion step for CSP. For **w/o similarity-based retrieval**, we remove the retrieval-time similarity constraints on anonymized oxidation-state formulas, elemental distances, and Wyckoff prototypes when selecting example structures. In this setting, we observe a clear degradation in both Match Rate and RMSE, indicating that structurally similar examples are crucial for steering the model toward high-quality, constraint-satisfying crystal structures.
> >
> > Regarding the requested algorithmic breakdown of the Retriever stage, we have **added a step-by-step algorithm description in the appendix** of the revised manuscript.
> >
> > **Question 4 -- Up until Section 4, the authors refer to the database as the data source used by the proposed framework. However, even from Section 4.1 it is not entirely clear what exactly is that database. Could the authors clarify?**
> >
> > **Response:** We apologize for the lack of clarity in the manuscript. The **"database" referred to** throughout the paper corresponds to **the training portion of the datasets** used in our experiments. Specifically, we **evaluate** CrystalAgent **on three datasets**: **MP20**, **MP52**, and the **Challenge Set** (Antunes et al., 2024). For each of these datasets, the training split is used as the retrieval database from which CrystalAgent obtains example structures during the retrieval stage. No additional external databases are introduced beyond these training splits.

---

> > > ### Author Response · Authors · 2025-12-02
> > > **Response to Reviewer 3mDi (3/7)**
> > >
> > > **Question 5 -- “The Challenge Set (Antunes et al., 2024) consists of 70 crystalline compounds… we evaluate on the 58 compounds from recent literature.”
> > > Was the framework validated using these 58 compounds? How is it guaranteed they are absent from the training data? Which training data?**
> > >
> > > **Response:** Yes. For the CSP experiments on the ChallengeSet, our evaluation is indeed conducted on the 58 compounds reported in recent literature.
> > >
> > > Regarding the reviewer’s questions about data overlap, the ChallengeSet (Antunes et al., 2024) and **its corresponding training split were constructed by the CrystaLLM authors**. We use these datasets exactly as released, without modification. The guarantee that the 58 evaluation compounds are absent from the training data is therefore inherited from the original dataset construction: in the CrystaLLM release, the ChallengeSet is explicitly separated from the training split, and the 58 evaluated compounds do not appear in the training portion of the dataset. Our framework simply uses the provided training split as the retrieval database and the provided ChallengeSet compounds as the test cases.
> > >
> > > When testing on the Challenge Set, the training data used as the retrieval database comes directly from the dataset assembled by (Antunes et al., 2024), which contains 2,047,889 CIF files collected from the Materials Project, OQMD, and NOMAD. According to the authors of (Antunes et al., 2024), the 58 ChallengeSet compounds taken from recent literature are explicitly guaranteed to be absent from this training split. We therefore rely on their construction to ensure that no overlap exists between the retrieval database and the evaluation compounds.
> > >
> > >
> > > > [1] Antunes, Luis M., Keith T. Butler, and Ricardo Grau-Crespo. "Crystal structure generation with autoregressive large language modeling." Nature Communications 15, no. 1 (2024): 10570.
> > >
> > > **Question 6 -- The test set of 58 compounds seems unreasonably small. Could the authors explain this design choice in detail, including aspects of generalizability, representativeness, and bias in evaluation metrics?**
> > >
> > > **Response:** The 58-compound test set is not constructed by us but is part of the ChallengeSet introduced by (Antunes et al., 2024). In our work, it is used **to assess crystal structure prediction performance on recently reported crystalline compounds** that are guaranteed to be absent from the training split. The ChallengeSet was intentionally designed as a manageable collection that spans a variety of solid-state structural classes, providing a fine-grained and representative evaluation of a model’s capabilities rather than a large-scale benchmark.
> > >
> > > > Moreover, we assemble what we call a challenge set, which consists of 70 structures, 58 of which were obtained from the recent literature, and were not in the training set. The remaining 12 structures are from the training set, and are included as representatives of different structural classes. They serve to assess the model’s ability to recover what it has seen in training, and as a means of comparing the model’s generations of seen and unseen structures. (Supplementary Table 1 contains the full list of the challenge set compounds, and their sources.) The permutative nature of the dataset, with many structures having been derived by substituting atoms into pre-defined templates, results in a test set with the potential for some structures to closely resemble those of the training set. The challenge set provides a source of structures that are guaranteed to have been produced through a different process. **Moreover, the challenge set constitutes a manageable set of compounds that reflects a variety of solid-state structural classes, allowing for a fine-grained picture of the model’s capabilities.**
> > > >
> > > > [1] Antunes, Luis M., Keith T. Butler, and Ricardo Grau-Crespo. "Crystal structure generation with autoregressive large language modeling." Nature Communications 15, no. 1 (2024): 10570.

---

> > > > ### Author Response · Authors · 2025-12-02
> > > > **Response to Reviewer 3mDi (4/7)**
> > > >
> > > > **Question 7 -- Why is the set of baselines different across Tables 1 and 2?**
> > > >
> > > > **Response:** The baselines differ between Tables 1 and 2 **because the two tables correspond to different tasks**. Table 1 evaluates crystal structure generation (CSG), which focuses on unconditional generation of crystal structures, whereas Table 2 evaluates crystal structure prediction (CSP), which requires generating a target structure conditioned on a given chemical formula. We follow the original papers when assembling the baselines: **FlowMM, CrystalTextLLM, MatExpert, and FLowLLM report results only for the CSG task** and therefore appear exclusively in Table 1, while **CrystaLLM and Mat2Seq provide results only for the CSP task** and therefore appear exclusively in Table 2.
> > > >
> > > > **Question 8 -- Why don’t the authors present error bars or deviations for any evaluations?**
> > > >
> > > > **Response:** We appreciate the reviewer’s question and address it as follows. We do not report error bars for two practical reasons. First, each evaluation involves a large number of test structures and requires repeated calls to computationally expensive relaxation tools as well as extensive LLM inference. The token consumption of the agentic workflow is substantial, and running multiple independent trials would incur prohibitive computational cost. Second, the baseline results are taken directly from prior publications, where error bars or deviation analyses are generally not provided. Reporting error bars only for CrystalAgent would therefore make the comparison inconsistent and potentially misleading.
> > > >
> > > > **Question 9 -- In my view, Table 1 does not present convincing evidence of the effectiveness of CrystalAgent in the CSG task. Structural Validity Rate and Coverage metrics are not informative since all methods score similarly. For the other metrics, CrystalAgent is somewhat better than CrystalTextLLM, FLowLLM, and CDVAE in Composition Validity Rate, but significantly worse otherwise. The authors advocate that another advantage of CrystalAgent is that it does not require fine-tuning, but I don’t see this as a serious argument in this context. Can the authors provide additional experimental results to support their claims?**
> > > >
> > > > **Response:** We thank the reviewer for the detailed feedback. We agree that, on the CSG task in Table 1, many metrics are close across methods: Structural Validity Rate and Coverage are almost saturated, and the remaining metrics show only modest numerical differences, even though CrystalAgent is competitive or slightly better in some of them. **To provide additional evidence for the effectiveness of our framework** and to illustrate **its reliance on only a small number of similar examples**, we therefore conduct an extra experiment on the CSP task using the MP20 dataset.
> > > >
> > > > Following the standard split used in prior work, we partition MP20 into train/valid/test subsets with 27,136 / 9,047 / 9,046 samples, respectively. To test how much CrystalAgent depends on the size of the retrieval database, we strongly subsample the training set. Concretely, we group all training structures by three keys simultaneously: (1) the anonymized oxidation-state based formula, (2) the space group, and (3) the Wyckoff prototype. We then discard all groups with fewer than three samples, and **for each remaining group we randomly keep only 3 crystal structures**. This yields 5,088 training structures in total, which is about 18.75% of the original training set. During evaluation, retrieval is restricted to this reduced training pool, while we still evaluate on the original MP20 test set.
> > > >
> > > > |                           |    MP20    |       |
> > > > | :-----------------------: | :--------: | :---: |
> > > > |                           | Match Rate | RMSE  |
> > > > |           CDVAE           |   33.9%    | 0.105 |
> > > > |          DiffCSP          |   51.5%    | 0.065 |
> > > > |         CrystaLLM         |   58.7%    | 0.041 |
> > > > |          Mat2Seq          |   61.3%    | 0.040 |
> > > > | Ours(18.75% retrieval DB) |   52.6%    | 0.033 |
> > > > |  Ours(full retrieval DB)  |   64.2%    | 0.034 |
> > > >
> > > > The results of this experiment are reported in the table above. Even when the retrieval database is reduced to less than one-fifth of its original size and each local neighborhood contains at most three examples, CrystalAgent maintains strong performance on CSP and remains competitive with or better than some of the baselines. This supports our claim that the proposed agent framework can effectively leverage a small number of structurally similar examples, rather than requiring a large, densely covered training set or extensive fine-tuning of the backbone model.

---

> > > > > ### Author Response · Authors · 2025-12-02
> > > > > **Response to Reviewer 3mDi (5/7)**
> > > > >
> > > > > **Question 10 -- Table 2 is suggesting that CrystalAgent is superior to other methods, however, there are multiple questions to this comparative analysis. Why is the Challenge Set evaluation limited to a single baseline, i.e. CrystaLLM? This appears as a major limitation. Also, to my knowledge, there are at least two versions of CrystaLLM – which one was used?**
> > > > >
> > > > > **Response:** The comparison on the Challenge Set is limited to CrystaLLM because we extract all baseline results directly from the original publications. As the authors of CrystaLLM introduced the Challenge Set, their paper is the only one that reports evaluation results on this benchmark. Earlier methods such as CDVAE and DiffCSP do not provide Challenge Set results, and although Mat2Seq follows the CrystaLLM framework, its paper does not report experiments on the Challenge Set either. Consequently, CrystaLLM is the only available baseline for this evaluation.
> > > > >
> > > > > Regarding **the version of CrystaLLM** used, we follow the results reported for **the large model architecture** trained only on the benchmark training sets, as this variant achieves the strongest performance on MP20 and MP52. Using the best-performing configuration ensures a fairer and more meaningful comparison against our framework.
> > > > >
> > > > > **Question 11 -- Table 3 shows promising results, but I find no data assets, repositories, or artifacts supporting this. Can the authors provide them?**
> > > > >
> > > > > **Response:** We will make all relevant materials available. Specifically, we will release an anonymous repository containing the code. We will provide an anonymized link to this anonymous repository in the rebuttal to support reproducibility.
> > > > >
> > > > > **Question 12 -- More generally, do the authors intend to release code, data assets, or artifacts to support reproducibility? Why aren’t those included in the submission?**
> > > > >
> > > > > **Response:** Yes, we intend to release all code required for reproducibility. The ICLR 2026 Author Guide (Q&A section) outlines three recommended options for anonymous code sharing, and we will follow the third option: once the discussion forum opens, we will post a comment addressed to the reviewers and area chairs containing a link to an anonymous repository. This approach makes the code accessible for evaluation while preserving anonymity.
> > > > >
> > > > > > Q: How can we make our code available for reviewing anonymously?
> > > > > > You can share your code in three ways:
> > > > > > 1. Anonymize your code, put it in a .zip file and submit it as supplementary materials.
> > > > > > 2. Make an anonymous repository and put the link in your paper. The above methods will make your code public, along with your paper and reviews/comments for the paper.
> > > > > > 3. After we open the discussion forums for all submitted papers, make a comment directed to the reviewers and area chairs and put a link to an anonymous repository. This method will let you keep your code visible only to the reviewers and ACs for your paper.
> > > > >
> > > > > **Question 13 -- Were only in silico models and tools used for validation?**
> > > > >
> > > > > **Response:** Yes. All validation in our work is performed using in silico models and computational tools. We rely exclusively on structure relaxation, energy evaluation, and symmetry or validity checks provided by established computational packages, and no experimental or laboratory-based validation is conducted. We have explicitly noted this point in the revised manuscript as part of the expanded discussion of limitations mentioned in our response to Question 2.

---

> > > > > > ### Author Response · Authors · 2025-12-02
> > > > > > **Response to Reviewer 3mDi (6/7)**
> > > > > >
> > > > > > **Question 14 -- The paper omits some recent and relevant works, e.g. SymmCD, DiffCSP++, deCIFer. Could the authors discuss how CrystalAgent compares to these methods conceptually and experimentally?**
> > > > > >
> > > > > > **Response:** We thank the reviewer for highlighting these recent and relevant works. Below, we discuss how CrystalAgent relates to SymmCD, DiffCSP++, and deCIFer, both conceptually and in terms of reported performance.
> > > > > >
> > > > > > Conceptually, SymmCD and DiffCSP++ are diffusion-based generative models that explicitly encode crystallographic symmetry into the generative process. SymmCD learns a joint distribution over an asymmetric unit and the associated symmetry transformations, and uses a diffusion model to generate symmetry-consistent crystals. DiffCSP++ extends DiffCSP by incorporating explicit space-group constraints into an equivariant diffusion process. deCIFer is an autoregressive language model that performs CSP conditioned on powder X-ray diffraction (PXRD) data and generates CIFs. In contrast, CrystalAgent wraps a general-purpose LLM in a constraint-aware, tool-augmented workflow that performs constraint completion, retrieval of similar examples, and tool-based validation/optimization, using in-context learning from a small set of retrieved crystal structures to infer atomic distribution patterns and generate new structures consistent with those patterns.
> > > > > >
> > > > > > |                | Validity Rate↑ |             | Coverage↑ |           | Distribution↓ |         |
> > > > > > | :------------: | :------------: | :---------: | :-------: | :-------: | :-----------: | :-----: |
> > > > > > |                |   Structural   | Composition |  Recall   | Precision |    Density    | Element |
> > > > > > |     CDVAE      |    100.00%     |   86.70%    |  99.15%   |  99.49%   |     0.688     |  0.278  |
> > > > > > |    DiffCSP     |    100.00%     |   83.25%    |  99.71%   |  99.76%   |     0.350     |  0.125  |
> > > > > > |     FlowMM     |     96.85%     |   83.19%    |  99.49%   |  99.58%   |     0.239     |  0.083  |
> > > > > > | CrystalTextLLM |     99.60%     |   95.40%    |  85.80%   |  98.90%   |     0.810     |  0.440  |
> > > > > > |   MatExpert    |     99.80%     |   96.10%    |  98.60%   |  99.10%   |     0.180     |  0.040  |
> > > > > > |    FLowLLM     |     99.94%     |   90.84%    |  96.95%   |  99.82%   |     1.140     |  0.150  |
> > > > > > |   DiffCSP++    |     99.94%     |   85.12%    |  99.73%   |  99.59%   |     0.235     |  0.375  |
> > > > > > |     SymmCD     |     90.34%     |   85.81%    |  99.58%   |  97.76%   |     0.230     |  0.400  |
> > > > > > |      Ours      |     99.77%     |   97.42%    |  98.93%   |  99.68%   |     0.339     |  0.141  |
> > > > > >
> > > > > > |           |   MP-20    |       |  MPTS-52   |       |
> > > > > > | :-------: | :--------: | :---: | :--------: | :---: |
> > > > > > |           | Match Rate | RMSE  | Match Rate | RMSE  |
> > > > > > |   CDVAE   |   33.9%    | 0.105 |    5.3%    | 0.211 |
> > > > > > |  DiffCSP  |   51.5%    | 0.065 |   12.2%    | 0.179 |
> > > > > > | CrystaLLM |   58.7%    | 0.041 |   19.2%    | 0.111 |
> > > > > > |  Mat2Seq  |   61.3%    | 0.040 |   23.1%    | 0.109 |
> > > > > > |  deCIFer  |   43.5%    | 0.076 |   11.4%    | 0.135 |
> > > > > > |   Ours    |   64.2%    | 0.034 |   31.0%    | 0.068 |
> > > > > >
> > > > > > Experimentally, SymmCD and DiffCSP++ primarily report results on crystal structure generation (CSG), so we incorporate their reported results into our CSG comparisons. For deCIFer, we include it in our CSP comparisons based on the metrics reported in its original work. Across both CSP and CSG settings, adding SymmCD, DiffCSP++, and deCIFer does not change our main conclusions: their performance falls within the range of baselines we already consider, and CrystalAgent continues to match or improve upon the existing baselines under our evaluation protocol.

---

> > > > > > > ### Author Response · Authors · 2025-12-02
> > > > > > > **Response to Reviewer 3mDi (7/7)**
> > > > > > >
> > > > > > > **Question 15 -- The authors claim that CrystalAgent bridges the gap caused by data scarcity, but the framework relies on a single database with the same limitations. How is this gap addressed in practice?**
> > > > > > >
> > > > > > > **Response:** Our claim does not refer to eliminating the need for a database, but to reducing the dependence on large, task-specific training datasets. **Although CrystalAgent uses a database for retrieval, this database is fully interchangeable and can be replaced or augmented at any time without retraining the model.** In practice, **the framework can operate with only a small number of representative examples**: a few retrieved structures are sufficient to guide the generation process toward the desired crystal prototypes. This stands in contrast to existing generative models, which require substantial supervised training on large curated datasets. *CrystalAgent therefore mitigates the impact of data scarcity by enabling crystal generation and design with minimal example data and without any additional model training.*
> > > > > > >
> > > > > > > **Question 16 -- Can properties of experimental synthesis or synthetic accessibility by itself be important constraints for the CSP and CSD tasks? Why aren’t these considered or discussed?**
> > > > > > >
> > > > > > > **Response:** We agree with the reviewer that experimental synthesis considerations and synthetic accessibility are important factors for CSP and CSD in real materials discovery workflows. These aspects are not incorporated into the current framework because CrystalAgent focuses specifically on structural generation and local relaxation using computational tools, and does not integrate thermodynamic or synthesis-related predictors. Incorporating such constraints would require additional models, datasets, or domain-specific heuristics that go beyond the scope of the present study. We recognize their importance, and we have explicitly discussed this omission as a limitation in the revised manuscript.
> > > > > > >
> > > > > > > **Question 17 -- The prompts in Appendix appear quite simple. Did the authors use prompt optimization tools or techniques? Please describe your approach to prompt and context engineering.**
> > > > > > >
> > > > > > > **Response:** No, we did not use any prompt optimization tools or automated prompt-search techniques. Our goal was to design a framework whose behavior is driven primarily by the agentic workflow and tool-based reasoning rather than by heavily engineered prompts. The prompts shown in the Appendix are intentionally simple because they serve only as high-level instructions that guide the LLM through retrieval, generation, and validation steps. The actual effectiveness of CrystalAgent comes from the structured use of external tools and the constraints enforced during retrieval and generation, not from prompt complexity.

---

### Official Review · Reviewer_ZHcf · 2025-11-01

**Soundness:** 2
**Presentation:** 3
**Contribution:** 2
**Rating:** 4
**Confidence:** 4

**Summary:**

This paper proposes CrystalAgent, a four-stage, training-free LLM agent for crystal structure generation and related tasks (CSG, CSP, and constrained CSD). The system decomposes the problem into (1) constraint extraction from user input, (2) retrieval and constraint completion from an in-domain crystal database (MP-20/MPTS-like), (3) in-context, constraint-aware structure generation with a general LLM, and (4) tool-based optimization/validation of candidates. On MP-20, MPTS-52, and a recent challenge set, the method reports very high validity and coverage and slightly higher match rates than prior LLM-based systems, while claiming not to fine-tune the base model.

**Strengths:**

1. The 4-stage workflow (extract $\rightarrow$ retrieve/complete $\rightarrow$ generate/check $\rightarrow$ optimize) is easy to follow and sits well within the current agentic AI for science. It shows how to make a general LLM act safely on domain-specific, highly structured outputs (crystals).

2. Using retrieval to fill in missing space groups / Wyckoff prototypes and then conditioning generation on them is a realistic design for real users who cannot specify full crystal details.

**Weaknesses:**

1. The proposed method runs on GPT-4o-mini, whereas several LLM baselines (CrystalTextLLM, MatExpert, and earlier CrystaLLM/Mat2Seq) rely on older/smaller backbones (LLaMA-2/3 or GPT-2–level). Part of the gain may therefore come from a much stronger, more recent model rather than from the agentic pipeline itself; this is a confound and makes the current comparison only partially fair.

2. Recent work MatLLMSearch [1]: Crystal Structure Discovery with Evolution-Guided LLMs follows the same high-level philosophy (frozen LLM + external loop/evaluator for crystal discovery) but is not compared or even discussed, so it is hard to see whether CrystalAgent is more sample-/compute-efficient than an evolutionary outer loop.

3. The method relies heavily on in-domain retrieval from MP-20 to provide in-context examples and to complete constraints. This reliance blurs the line between “no training” and “using the train set as a memory,” and raises the question of how well the method works on genuinely out-of-distribution sources.

4. Most evaluations use surrogate/ML potentials and structural metrics; there is little DFT-level or experimental validation to show that the generated candidates are actually useful to materials scientists.

5. With several moving parts (constraint completion, similarity-based retrieval, generation-time checks, optimization), the paper needs clearer ablations to isolate which components actually drive the improvements.

[1] Gan J, Zhong P, Du Y, et al. Large language models are innate crystal structure generators[C]//AI for Accelerated Materials Design-ICLR 2025. 2025.

**Questions:**

1. Can you report CrystalAgent with a matched backbone (e.g., LLaMA-2-13B / LLaMA-3-70B) to show that the pipeline itself, not GPT-4o-mini, is responsible for the improvements? Alternatively, can you re-run at least one LLM baseline under GPT-4o-mini with your retrieval + check settings?

2. How does CrystalAgent compare to MatLLMSearch under the same evaluator (same ML potential / same DFT budget) in terms of validity, metastable/stable rate, and candidate diversity? What is the compute cost difference between your fixed 4-stage pipeline and their evolutionary loop?

3. Since your retrieval pool is MP-20-like and your evaluation is also on MP-20/MPTS, what happens if the reference database is swapped to another source (e.g., OQMD/ICSD subset) or you evaluate on a held-out recent literature set without close neighbors?

4. What is the fraction of generations that pass your constraint checks on the first attempt, and how many re-generations are typically needed per final valid structure? This matters for real-time agentic usage.

5. Which structure optimization/energy evaluation tools are actually used in stage 4, and do their training data overlap with MP-20? Can you clarify the cost per candidate and whether this step ever rejects structures that pass stage 3?

---

> ### Author Response · Authors · 2025-12-02
> **Response to Reviewer ZHcf (1/5)**
>
> We would like to address the weaknesses and questions you pointed out one by one below.
>
> **Weakness 1 -- The proposed method runs on GPT-4o-mini, whereas several LLM baselines (CrystalTextLLM, MatExpert, and earlier CrystaLLM/Mat2Seq) rely on older/smaller backbones (LLaMA-2/3 or GPT-2–level). Part of the gain may therefore come from a much stronger, more recent model rather than from the agentic pipeline itself; this is a confound and makes the current comparison only partially fair.**
>
> **Response:** We agree that backbone strength can be a potential confound, and we have therefore added experiments using LLaMA-3 70B in our method. The comparative results are reported in the table below. As shown there, replacing GPT-4o-mini with LLaMA-3 70B leads to only minor fluctuations in performance, and our method continues to outperform the baselines by a similar margin. This suggests that the gains primarily stem from the proposed agentic pipeline rather than from a particular choice of backbone.
>
> |                   |   MP-20    |       |  MPTS-52   |       |
> | :---------------: | :--------: | :---: | :--------: | :---: |
> |                   | Match Rate | RMSE  | Match Rate | RMSE  |
> |       CDVAE       |   33.9%    | 0.105 |    5.3%    | 0.211 |
> |      DiffCSP      |   51.5%    | 0.065 |   12.2%    | 0.179 |
> |     CrystaLLM     |   58.7%    | 0.041 |   19.2%    | 0.111 |
> |      Mat2Seq      |   61.3%    | 0.040 |   23.1%    | 0.109 |
> | Ours(Llama3-70B)  |   65.1%    | 0.034 |   32.0%    | 0.067 |
> | Ours(GPT-4o-mini) |   64.2%    | 0.034 |   31.0%    | 0.068 |
>
> More specifically, because the current Ollama's LLaMA-3 does not support the tool-calling interface we rely on to strictly constrain output formats, we only swap the Generation component to LLaMA-3 70B. The structured-output components of our pipeline (e.g., crystal-constraint extraction) still use GPT-4o-mini with tool calls. This setup isolates the effect of changing the generative backbone while keeping the downstream structured reasoning unchanged, and the stability of results across GPT-4o-mini and LLaMA-3 70B further supports that our improvements are not an artifact of a stronger single model.
>
> **Weakness 2 -- Recent work MatLLMSearch [1]: Crystal Structure Discovery with Evolution-Guided LLMs follows the same high-level philosophy (frozen LLM + external loop/evaluator for crystal discovery) but is not compared or even discussed, so it is hard to see whether CrystalAgent is more sample-/compute-efficient than an evolutionary outer loop.**
>
> **Response:** We thank the reviewer for raising this point. **MatLLMSearch has already been discussed in the Related Work section of our manuscript.** It follows a similar high-level idea of using a frozen LLM with an external evaluation loop for crystal discovery, but its concrete procedure and computational cost are quite different from CrystalAgent.
>
> MatLLMSearch adopts an evolutionary outer loop. For a given CSP target (e.g., a fixed composition such as $\text{Ag}_6\text{O}_2$), it runs multiple evolution rounds and, in each round, generates a large population of candidate structures via repeated LLM calls, followed by ranking and refinement with CHGNet. As described in **MatLLMSearch** and confirmed by our reproduction, this typically requires about **10 evolution rounds** with roughly **100 LLM calls per round** for a single composition. Each LLM call takes about 10–25 seconds, so a **single evolution round** already **takes around 25 minutes** just for the 100 LLM calls. The subsequent **CHGNet** relaxations **further increase the total runtime**.
>
> In contrast, **CrystalAgent** is designed for much lighter per-query usage. For a given chemical formula or constraint set, our framework performs 1 retrieval over the database and at most 3 LLM calls for structure generation, controlled by constraint checks. Each final candidate structure is relaxed once by M3GNet. In our implementation, **the average wall-clock time per composition is under 30 seconds**. These observations indicate that CrystalAgent is substantially more sample- and compute-efficient for CSP-style queries than an evolutionary outer loop such as MatLLMSearch.

---

> > ### Author Response · Authors · 2025-12-02
> > **Response to Reviewer ZHcf (2/5)**
> >
> > **Weakness 3 -- The method relies heavily on in-domain retrieval from MP-20 to provide in-context examples and to complete constraints. This reliance blurs the line between “no training” and “using the train set as a memory,” and raises the question of how well the method works on genuinely out-of-distribution sources.**
> >
> > **Response:** We would like to clarify that **we do not use MP-20 alone as the retrieval database**. In our experiments **we evaluate on three datasets: MP-20, MPTS-52, and the CrystaLLM's Challenge set**. For each of these benchmarks, **we use the corresponding training split as the retrieval set** and evaluate only on the test split, following the dataset splits defined in prior work. Under this protocol, **the retrieval set and the test set are strictly disjoint**, so no test structure can ever be retrieved as context. Therefore, **our method does not treat the train set as a memory** of test examples, but instead uses the standard setting of retrieving from the training split while evaluating on a held-out test split. **Our claim of “no additional model training” is meant in the narrow sense that we do not perform any further gradient-based updates or fine-tuning of model parameters beyond the backbone models used in prior work; the training splits are only used as a retrieval corpus at inference time.**
> >
> > Regarding the question of how the method behaves on genuinely out-of-distribution sources, **our current formulation in fact does not target such OOD settings**. **The method is explicitly designed to rely on structurally similar crystals as references.** In particular, the agent only generates a candidate structure when it can retrieve example crystals that share the same anonymized chemical formula, and the same space group as the target. If no such structurally compatible examples are found in the database, the system simply reports that no suitable reference is available and does not attempt to generate a structure. Consequently, the scenarios we consider in this work do not involve truly out-of-distribution cases by design; extending the approach to settings without any suitable matches is an interesting direction for future work.
> >
> > **In the paper, we also provide experiments that relate to the out-of-distribution question by evaluating on the Challenge Set introduced by CrystaLLM**, which is specifically designed to test models on compounds absent from the training data. In this setting, the training data used as the retrieval database consists of 2,047,889 CIF files collected from the Materials Project, OQMD, and NOMAD, and the evaluation is carried out on 58 crystalline compounds from recent literature that are explicitly guaranteed to be absent from this training split. We use these datasets exactly as released in the CrystaLLM work, that is, we use the provided training split as the retrieval corpus and the 58 Challenge Set compounds as test cases, ensuring that there is no overlap between the retrieval database and the evaluated structures. On this OOD-style benchmark, our method consistently outperforms CrystaLLM (see Table below), which indicates that the proposed agentic pipeline maintains strong performance even on compounds that lie outside the specific Materials Project distributions, as long as structurally related prototypes exist in the underlying training corpus.
> >
> > |           | Challenge Set |       |
> > | :-------: | :-----------: | :---: |
> > |           |  Match Rate   | RMSE  |
> > | CrystaLLM |     22.4%     | 0.090 |
> > |   Ours    |     32.8%     | 0.096 |
> >
> > **Weakness 4 -- Most evaluations use surrogate/ML potentials and structural metrics; there is little DFT-level or experimental validation to show that the generated candidates are actually useful to materials scientists.**
> >
> > **Response:** We agree that ultimately DFT-level and experimental validation are crucial for demonstrating the practical usefulness of generated crystal structures to materials scientists. In the present work, we focus on methodological development and on systematic comparisons under standard surrogate metrics and ML-based stability predictors that are widely adopted in prior literature. A comprehensive DFT or experimental validation campaign for a large number of generated candidates is beyond the scope of this study and remains an important direction for future work. In particular, integrating our agentic pipeline into closed-loop workflows with high-fidelity DFT calculations and, eventually, targeted experimental synthesis will be a key step toward fully assessing and realizing the practical impact of the proposed method.

---

> > > ### Author Response · Authors · 2025-12-02
> > > **Response to Reviewer ZHcf (3/5)**
> > >
> > > **Weakness 5 -- With several moving parts (constraint completion, similarity-based retrieval, generation-time checks, optimization), the paper needs clearer ablations to isolate which components actually drive the improvements.**
> > >
> > > **Response:** We appreciate the reviewer’s suggestion and have conducted an ablation study on the MP20 dataset for the CSP task to quantify the contribution of each component in the table below.
> > >
> > > |                                |    MP20    |       |
> > > | :----------------------------: | :--------: | :---: |
> > > |                                | Match Rate | RMSE  |
> > > |   w/o constraint completion    |   42.8%    | 0.035 |
> > > | w/o similarity-based retrieval |   57.8%    | 0.043 |
> > > |   w/o generation-time checks   |   64.2%    | 0.035 |
> > > |        w/o optimization        |   62.9%    | 0.054 |
> > > |              Ours              |   64.2%    | 0.034 |
> > >
> > > For **w/o constraint completion**, we replace the space-group prediction tool used to complete missing symmetry constraints from the given composition with a simple prior based on the distribution of space groups for anonymized compositions in existing retrieval databases. This leads to a substantial drop in Match Rate, highlighting the importance of an accurate constraint-completion step for CSP. For **w/o similarity-based retrieval**, we remove the retrieval-time similarity constraints on anonymized oxidation-state formulas, elemental distances, and Wyckoff prototypes when selecting example structures. In this setting, we observe a clear degradation in both Match Rate and RMSE, indicating that structurally similar examples are crucial for steering the model toward high-quality, constraint-satisfying crystal structures. For **w/o generation-time checks**, we disable the validity checks on the generated structures. The performance remains nearly unchanged, indicating that our crystal-structure representation and constraints already lead the model to generate structures that satisfy the required constraints. Finally, for **w/o optimization**, we remove the post-generation optimization stage. This variant yields a slightly lower Match Rate and a higher RMSE, indicating that, while the large language model can propose reasonable initial candidates, an explicit optimization module is still necessary to refine these candidates and approach potentially optimal crystal structures.
> > >
> > >
> > > **Question 1 -- Can you report CrystalAgent with a matched backbone (e.g., LLaMA-2-13B / LLaMA-3-70B) to show that the pipeline itself, not GPT-4o-mini, is responsible for the improvements? Alternatively, can you re-run at least one LLM baseline under GPT-4o-mini with your retrieval + check settings?**
> > >
> > > **Response:** We have addressed this concern by adding a matched-backbone variant of our method that uses LLaMA-3 70B as the generative backbone. The results are reported in the table below. As shown there, the LLaMA-3 70B variant of our method achieves almost the same performance as the GPT-4o-mini variant on both MP-20 and MPTS-52, and both variants maintain a similar margin over all baselines. This indicates that the performance gains are largely attributable to the proposed retrieval and checking pipeline rather than to a specific backbone choice. Because the current LLaMA-3 API does not support the tool-calling interface that we use to enforce structured outputs, we only replace the generation component with LLaMA-3 70B, while keeping the constraint extraction and other structured-output components based on GPT-4o-mini. This setup isolates the effect of changing the generative backbone and shows that our pipeline is robust across strong modern LLM backbones.
> > >
> > > |                   |   MP-20    |       |  MPTS-52   |       |
> > > | :---------------: | :--------: | :---: | :--------: | :---: |
> > > |                   | Match Rate | RMSE  | Match Rate | RMSE  |
> > > |       CDVAE       |   33.9%    | 0.105 |    5.3%    | 0.211 |
> > > |      DiffCSP      |   51.5%    | 0.065 |   12.2%    | 0.179 |
> > > |     CrystaLLM     |   58.7%    | 0.041 |   19.2%    | 0.111 |
> > > |      Mat2Seq      |   61.3%    | 0.040 |   23.1%    | 0.109 |
> > > | Ours(Llama3-70B)  |   65.1%    | 0.034 |   32.0%    | 0.067 |
> > > | Ours(GPT-4o-mini) |   64.2%    | 0.034 |   31.0%    | 0.068 |

---

> > > > ### Author Response · Authors · 2025-12-02
> > > > **Response to Reviewer ZHcf (4/5)**
> > > >
> > > > **Question 2 -- How does CrystalAgent compare to MatLLMSearch under the same evaluator (same ML potential / same DFT budget) in terms of validity, metastable/stable rate, and candidate diversity? What is the compute cost difference between your fixed 4-stage pipeline and their evolutionary loop?**
> > > >
> > > > **Response:** We thank the reviewer for this question and agree that a direct comparison under the same evaluator would be very informative. We have not yet reimplemented MatLLMSearch and CrystalAgent under the same ML potential or DFT budget, so we cannot provide a strictly fair numerical comparison in terms of validity, stable/metastable rates, or candidate diversity. Specifically, MatLLMSearch is implemented and evaluated on the Matbench-bandgap dataset, and its evolutionary search operates over a crystal pool that has been prefiltered in a task-specific way. This setup does not translate directly to our MP20-based CSG benchmarks, which makes a controlled reproduction under exactly the same conditions non-trivial.
> > > >
> > > > To at least compare computational cost on a CSP task, we follow the case study in **MatLLMSearch** for the composition $\text{Ag}_6\text{O}_2$, using GPT-4o-mini as the LLM backbone. As described in MatLLMSearch and confirmed by our reproduction, this typically requires about 10 evolution rounds with roughly 100 LLM calls per round for a single composition. Each LLM call takes about 10–25 seconds, so a **single evolution round already takes around 25 minutes just for the 100 LLM calls**. The subsequent CHGNet relaxations further increase the total runtime.
> > > >
> > > > **CrystalAgent**, by contrast, uses a fixed and lightweight four-stage pipeline. For a given chemical formula or constraint set, it performs 1 retrieval over the database, at most 3 LLM calls for structure generation, and a single M3GNet relaxation for each final candidate. In our implementation, **the average wall-clock time per composition is under 30 seconds**. While a fully controlled comparison under the same ML potential and DFT budget is left for future work, these observations indicate that our fixed pipeline is substantially more sample- and compute-efficient per CSP query than an evolutionary loop such as MatLLMSearch, and is better suited to real-time, interactive use.
> > > >
> > > > **Question 3 -- Since your retrieval pool is MP-20-like and your evaluation is also on MP-20/MPTS, what happens if the reference database is swapped to another source (e.g., OQMD/ICSD subset) or you evaluate on a held-out recent literature set without close neighbors?**
> > > >
> > > > **Response:** **Our retrieval pool is not restricted to MP-20.** In addition to MP-20 and MPTS-52, we also evaluate on the CrystaLLM Challenge Set, where the retrieval database is the large corpus assembled by Antunes et al. from the Materials Project, OQMD, and NOMAD, and the test structures are 58 recent-literature compounds that are guaranteed to be absent from this training split. This setting effectively corresponds to swapping the retrieval source to a broader MP/OQMD/NOMAD pool and evaluating on a held-out recent literature set, and on this benchmark our method still outperforms CrystaLLM (see table below). This suggests that the proposed agentic pipeline is robust to changes in the underlying retrieval database, as long as structurally related prototypes are available.
> > > >
> > > > |           | Challenge Set |       |
> > > > | :-------: | :-----------: | :---: |
> > > > |           |  Match Rate   | RMSE  |
> > > > | CrystaLLM |     22.4%     | 0.090 |
> > > > |   Ours    |     32.8%     | 0.096 |
> > > >
> > > > Conceptually, **the framework is database-agnostic: one can replace the retrieval pool with other curated sources such as OQMD or ICSD subsets without changing the pipeline itself**; the key requirement is that the database contains examples sharing the same anonymized chemical formula, space group, and Wyckoff prototype as the target. Systematically benchmarking across different retrieval corpora is an interesting direction for future work.
> > > >
> > > > **For the case where the evaluation compounds have no close neighbors in the reference database, our current method is designed to abstain.** If the retrieval stage fails to find structurally compatible examples under the above criteria, the agent explicitly reports that no suitable reference is available and does not attempt to generate a structure. As a result, the present work does not target regimes where no prototype-level neighbors exist; extending the approach to such genuinely neighbor-free settings is left as future work.

---

> > > > > ### Author Response · Authors · 2025-12-02
> > > > > **Response to Reviewer ZHcf (5/5)**
> > > > >
> > > > > **Question 4 -- What is the fraction of generations that pass your constraint checks on the first attempt, and how many re-generations are typically needed per final valid structure? This matters for real-time agentic usage.**
> > > > >
> > > > > **Response:** We evaluate this on the CSP task using the MP20 dataset. For computational efficiency, we randomly select 1000 samples from the MP20 test set and record the number of generation attempts required for the generated structure to pass the generation check. A count of 0 indicates that the MP20 training set (i.e., the retrieval database) does not contain any chemically similar crystal examples, so no generation attempt is performed. A count of 1 indicates that the structure passed the generation check on the first attempt. Counts of 2 and 3 correspond to one or two regenerations, respectively, with 3 being the maximum allowed number of attempts. Overall, the data in the table below show that for real-time applications, the overhead of repeated generation is limited, and the system can typically return a constraint-satisfying crystal structure after a single LLM call.
> > > > >
> > > > > | Generation Attempts | Sample Count | Percent |
> > > > > | :-----------------: | :----------: | :-----: |
> > > > > |          0          |      22      |  2.2%   |
> > > > > |          1          |     941      |  94.1%  |
> > > > > |          2          |      11      |  1.1%   |
> > > > > |          3          |      26      |  2.6%   |
> > > > >
> > > > >
> > > > > **Question 5 -- Which structure optimization/energy evaluation tools are actually used in stage 4, and do their training data overlap with MP-20? Can you clarify the cost per candidate and whether this step ever rejects structures that pass stage 3?**
> > > > >
> > > > > **Response:** In stage 4, we use the **MatGL implementation of the M3GNet** to perform structure relaxation and obtain approximate energies for the generated candidates. M3GNet is trained on a very large corpus of DFT relaxation trajectories `MPF.2021.2.8` from the Materials Project. In terms of computational cost, running M3GNet-based relaxation for a single candidate structure typically takes on the order of **10–30 seconds** in our experiments. **This step does not reject structures that have passed our stage-3 generation checks.** Instead, it is used as a post-processing refinement: if the relaxed structure preserves the target symmetry and Wyckoff prototype, we keep the relaxed version; if relaxation changes the space group or the Wyckoff prototype relative to the pre-relaxation structure, we reject the optimized structure and fall back to the original stage-3 candidate. Thus, stage 4 never filters out candidates that satisfy the generation constraints; it only decides whether to adopt the relaxed or the original version of the same candidate.

---

### Official Review · Reviewer_QbFC · 2025-11-01

**Soundness:** 1
**Presentation:** 4
**Contribution:** 1
**Rating:** 0
**Confidence:** 4

**Summary:**

The work proposes CrystalAgent, which is a framework for crystal structure generation that uses a large language model (LLM) in a four-stage pipeline: extracting constraints from natural language inputs, retrieving similar structures from a database, generating candidate structures via in-context learning, and optimizing them using external tools. The system achieves good performance on crystal structure generation, prediction and design (CSG, CSP, CSD) without task-specific fine-tuning.

**Strengths:**

1. The work uses symmetry-based encoding that represents crystals using space groups, lattice parameters, and Wyckoff positions rather than raw atomic coordinates. The choice is reasonable and physically sound.

2. The system can handle multiple tasks (CSG, CSP, CSD) without specific fine-tuning, given the advantage of tool use.

**Weaknesses:**

W1. The optimization step is fundamentally flawed. Comparing total energy is valid for comparing polymorphs of the same composition (CSP tasks only), but it does not apply to CSG or CSD. It is formation energy that you want to approach. Such described 'optimization' would eventually favor structures with certain patterns e.g. fewer atoms simply because they have lower total energy.
W2. The concept of 'thermodynamic stability' is abused in the draft. Overclaims like 'ensures that the final output is ... thermodynamically stable' were emphasized multiple times. Expressions like 'ensure structural stability and physical plausibility' are scientifically invalid. In addition, a local minima does not guarantee the structure is stable. Even an M3GNet relaxed crystal structure with negative formation energy only indicates the structure is metastable with respect to decomposition.
W3. The work is mispositioned as "agentic reasoning". The claimed "agentic reasoning" is a deterministic pipeline of retrieval, few-shot prompting, and static tool use for validation checks. It shows no sign that the designed system can benefit from autonomous, adaptive reasoning from a real agentic system.
W4. Instead of 'training-free', 'fine-tuning free' is more appropriate, as you concluded in the last paragraph.

**Questions:**

Q1. Is the optimization step shared by all tasks?

---

> ### Author Response · Authors · 2025-12-02
> **Response to Reviewer QbFC (1/2)**
>
> Although you remain unconvinced about the overall soundness and contributions of our work, we genuinely appreciate your positive comments on our presentation. We would like to address the weaknesses and questions you pointed out one by one below.
>
> **Weakness 1 -- The optimization step is fundamentally flawed. Comparing total energy is valid for comparing polymorphs of the same composition (CSP tasks only), but it does not apply to CSG or CSD. It is formation energy that you want to approach. Such described 'optimization' would eventually favor structures with certain patterns e.g. fewer atoms simply because they have lower total energy.**
>
> **Response:** We respectfully **disagree** with the statement that our optimization step applies only to CSP tasks and is not suitable for CSG or CSD tasks. In our framework, the optimization stage contains two components. First, each generated crystal structure is relaxed to obtain a locally optimized configuration. Second, among the structures of the **same composition generated from different Wyckoff prototypes**, we select the one that exhibits the lowest energy.
>
> The key point is that the **retrieval stage refines the chemical composition constraints** by leveraging the retrieved example structures. *As a result, the generation stage produces **candidate structures** that all **share the same composition** across **CSP, CSG, and CSD** settings.* With the composition fixed, **comparing the energy per atom becomes equivalent to comparing formation energies**, because the reference chemical potentials cancel out. Therefore, selecting the most stable structure among the candidates is well defined for all three tasks.
>
> We acknowledge a typographical oversight in the manuscript. *Our selection criterion should have been described as **the energy per atom rather than the total energy**.* We appreciate the reviewer for noticing this and **have corrected this in the revised version.**.
>
> **Weakness 2 -- The concept of 'thermodynamic stability' is abused in the draft. Overclaims like 'ensures that the final output is ... thermodynamically stable' were emphasized multiple times. Expressions like 'ensure structural stability and physical plausibility' are scientifically invalid. In addition, a local minima does not guarantee the structure is stable. Even an M3GNet relaxed crystal structure with negative formation energy only indicates the structure is metastable with respect to decomposition.**
>
> **Response:** We appreciate the reviewer's careful attention to our terminology. We agree that expressions such as "thermodynamically stable", "structurally stable", and similar formulations were not sufficiently precise in the current draft. Our intention was not to claim that the proposed framework can determine true thermodynamic stability or guarantee global stability of the generated structures. *Rather, **the optimization step** is intended only to **relax each candidate structure** of the same chemical composition to a nearby local minimum and to **select the lowest-energy structure among them**.*
>
> We fully agree that a relaxed structure in a local minimum, even with negative formation energy, may still be metastable with respect to decomposition. The current framework does not attempt to assess global thermodynamic stability, phase competition, or synthesizability, and we have revised the manuscript to ensure that none of these implications are inadvertently conveyed.
>
> In the revised version, we have replaced the inaccurate statements with more scientifically appropriate descriptions, such as "local relaxation" and "selecting the lowest-energy structure among candidates of the same composition", which more accurately reflect what the current system is designed to accomplish.

---

> > ### Author Response · Authors · 2025-12-02
> > **Response to Reviewer QbFC (2/2)**
> >
> > **Weakness 3 -- The work is mispositioned as "agentic reasoning". The claimed "agentic reasoning" is a deterministic pipeline of retrieval, few-shot prompting, and static tool use for validation checks. It shows no sign that the designed system can benefit from autonomous, adaptive reasoning from a real agentic system.**
> >
> > **Response:** We thank the reviewer for pointing this out and agree that our current wording around “agentic reasoning” may be misleading. While our system can automatically handle different crystal structure generation requirements and route inputs through different constraint-completion steps and tool invocations, it is still best characterized as a deterministic, multi-stage pipeline that combines constraint completion, retrieval, LLM-based generation, and tool-based validation and optimization, rather than a fully autonomous, adaptive agent in the broader sense. In the revised version, we have refined the terminology to emphasize that our method is a specialized crystal structure generation agent, rather than framing it as a general “agentic reasoning” system.
> >
> > **Weakness 4 -- Instead of 'training-free', 'fine-tuning free' is more appropriate, as you concluded in the last paragraph.**
> >
> > **Response:** We thank the reviewer for this helpful comment. We have rephrased “training-free” to “fine-tuning free” in the last paragraph of the revised manuscript.
> >
> > **Question 1 -- Is the optimization step shared by all tasks?**
> >
> > **Response:** **Yes**, the optimization step is indeed shared across CSP, CSG, and CSD tasks. As detailed in our response to Weakness 1, **this unified design is a direct consequence of how the chemical composition constraints are enforced during retrieval and generation**. The retrieval stage refines the chemical composition constraints using the retrieved example structures, **ensuring that all candidate structures entering the optimization stage share the same composition for all three tasks**. Under this condition, *the purpose of the energy-based ranking is to identify the lowest-energy structure among candidate crystal structures of the same composition, which in turn makes the optimization step applicable to CSP, CSG, and CSD alike.*

---

### Author Response · Authors · 2025-12-02
**Anonymous Code Repository**

Dear reviewers and area chairs,

We provide our anonymous code repository here: https://anonymous.4open.science/r/CrystalAgent-A7FD/

It contains the full implementation of CrystalAgent.

---

### Meta-Review · Area_Chair_bEDi · 2026-01-07

**Summary:**

The paper proposes a framework for crystal structure generation using LLMs. Reviewers however, expressed concerns on (1) flawed optimization step, (2) over claim of thermodynamic stability, (3) no real agentic reasoning capability (4) missing baselines in comparison, lack of ablation study (5) choice of small evaluation data set. The authors claim that they will revise the writing to, add additional baseline and ablation studies. However, this would lead to a major revision of the manuscript.

**Reviewer Concerns:**

All concerns are addressed or partially addressed.

**Reviewer Scores:**

Reviewers 3mDi and QbFC may consider to increase their scores.

---

### Decision · Program_Chairs · 2026-01-26

Reject